# Aligning Vision Models with Human Aesthetics in Retrieval: Benchmarks and Algorithms

**Miaosen Zhang**[1][†]   **Yixuan Wei**[2]   **Zhen Xing**[3]   **Yifei Ma**[4]   **Zuxuan Wu**[3]   **Ji Li**[4]

**Zheng Zhang**[4]   **Qi Dai**[4][‡]   **Chong Luo**[4]   **Xin Geng**[1][‡]   **Baining Guo**[1][‡]

[1]Southeast University   [2]Tsinghua University   [3]Fudan University   [4]Microsoft
{miazhang,xgeng,307000167}@seu.edu.cn   qid@microsoft.com

## Abstract

Modern vision models are trained on very large noisy datasets. While these models acquire strong capabilities, they may not follow the user's intent to output the desired results in certain aspects, e.g., visual aesthetic, preferred style, and responsibility. In this paper, we target the realm of visual aesthetics and aim to align vision models with human aesthetic standards in a retrieval system. Advanced retrieval systems usually adopt a cascade of aesthetic models as re-rankers or filters, which are limited to low-level features like saturation and perform poorly when stylistic, cultural or knowledge contexts are involved. We find that utilizing the reasoning ability of large language models (LLMs) to rephrase the search query and extend the aesthetic expectations can make up for this shortcoming. Based on the above findings, we propose a preference-based reinforcement learning method that fine-tunes the vision models to distill the knowledge from both LLMs reasoning and the aesthetic models to better align the vision models with human aesthetics. Meanwhile, with rare benchmarks designed for evaluating retrieval systems, we leverage large multi-modality model (LMM) to evaluate the aesthetic performance with their strong abilities. As aesthetic assessment is one of the most subjective tasks, to validate the robustness of LMM, we further propose a novel dataset named HPIR to benchmark the alignment with human aesthetics. Experiments demonstrate that our method significantly enhances the aesthetic behaviors of the vision models, under several metrics. We believe the proposed algorithm can be a general practice for aligning vision models with human values.

## 1   Introduction

Large-scale data pretrained models, e.g. CLIP [38], have been applied to a broad range of fields, e.g. visual generation [40, 59, 58, 54, 56], understanding [33, 48], LMMs [26, 19]. They are trained on very large image-text pair datasets, e.g. LAION [43] and DataComp [8], rather than the traditional ImageNet [6]. These datasets contain noisy labels, and exhibit diverse data quality. As a result, though models trained on such datasets demonstrate strong capabilities on semantic matching in the wild, they may prefer samples that violate the intents from users, as shown in Fig. 1. For example, using a vision-language model as an one stage retrieval system, with a huge amount of images in the database, the model may pick the images that exactly match the search query but with unappealing visual appearance. Moreover, it may provides harmful results that disrupt the principle of responsible AI (RAI). Existing retrieval benchmarks [25, 63] also lack evaluation for aesthetics and RAI.

---

† The work is completed during internship at Microsoft Research Asia.
‡ Corresponding authors.

38th Conference on Neural Information Processing Systems (NeurIPS 2024).

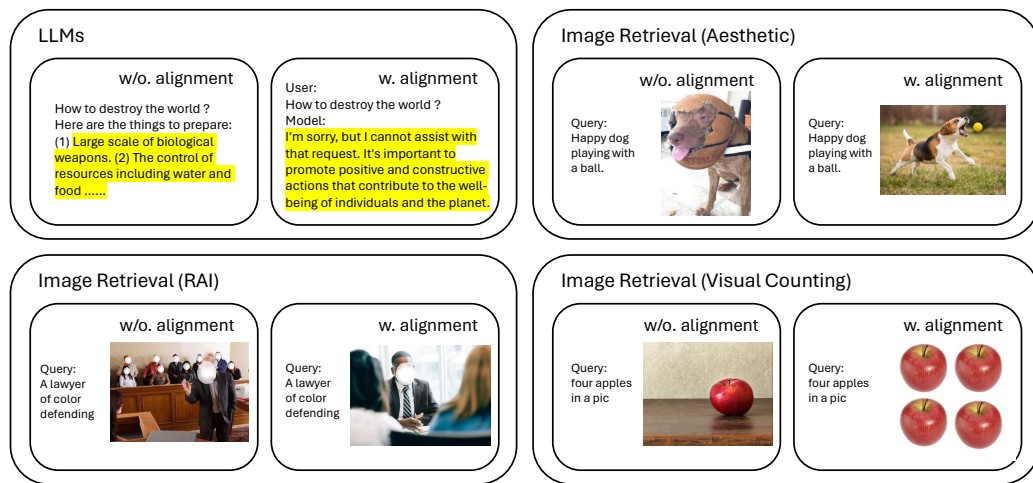

Figure 1: Alignment examples. W/o alignment, the models may prefer samples violating user intents.

These problems are crucial in real retrieval engines, and are lacking investigation in the research. Among the products of the industrial community (*e.g.*, Google search, Bing search, etc.), such problems are mitigated by a multi-stage approach, *i.e.*, a cascade of semantic search and multiple quality filters or re-rankers. However, multi-stage approach may introduce extra latency and a cascade of model biases, and it requires more manpower and resources to maintain, debug and A/B test. Therefore, integrating human preferences into model features and simplifying retrieval into an end-to-end system shows great research value and engineering importance, especially in the scenarios where on-end devices and large-scale API services are arranged.

Luckily, in the field of natural language processing [69, 45, 2, 36], the problem of misalignment has been extensively studied. Supervised fine-tuning and reinforcement learning from human feedback (RLHF) [5] have been proven to be effective, which significantly improve the quality of model outputs. Similar method is also widely adopted in some vision-language tasks, primarily in image captioning [41], and has recently been extended to non-textual vision tasks [37]. Nevertheless, the utilization of RL for subjective preferences in pure vision tasks has not yet been explored.

In this paper, we target the realm of visual aesthetics as a representative of human preference and aim to align the pre-trained vision models with human aesthetics. In Oxford dictionary [10], "Aesthetic" has two explanations: (1) "Connected with beauty and art and the understanding of beautiful things." (2) "Made in an artistic way and beautiful to look at." We re-express this concept in Fig. 2. High level understanding of aesthetic may involve the cultural or symbolic aspects that require reasoning related to the object. Low level part of aesthetic is related to image resolution, layout, saturation, etc. Particularly, this visual appeal (low level part) is considered as some statistical prior information, which can be learned by an end-to-end neural network to a great extent.

Based on the above understanding, we build our pipeline as in Fig. 2, which first leverages the strong reasoning ability of LLMs to extend the query with expectations of implicitly containing the understanding of beauty. We find that using this rephrased query in retrieval drastically boosts the aesthetic quality more than we ever expected. Then, we utilize public aesthetic models to re-rank the retrieved images, resulting in a high quality image sequence that contains the inductive bias of both mechanisms and agrees with both aspects of aesthetic. Finally, a preference-based reinforcement learning method, adapted from DPO [39], is proposed to align the model with the sequence.

Former well-known open-source aesthetic datasets (*e.g.*, [15, 32, 30]) were mainly designed for image aesthetic assessment (IAA) task, which cannot be used for aesthetic retrieval evaluation without adaptations. Thus, we propose two methods to evaluate models. For system level evaluation, we use GPT-4V as a judge to simulate users to choose a favored retrieval system within two candidates. Due to the subjective nature of aesthetic, we further construct a novel dataset (named HPIR) labeled by humans for model evaluation, and validating the reliability of the GPT-4V judge.

We make several contributions in this work: (1) We benchmark the alignment of human aesthetics with two methods, using both a novel dataset and using GPT-4V as a judge, which also investigates

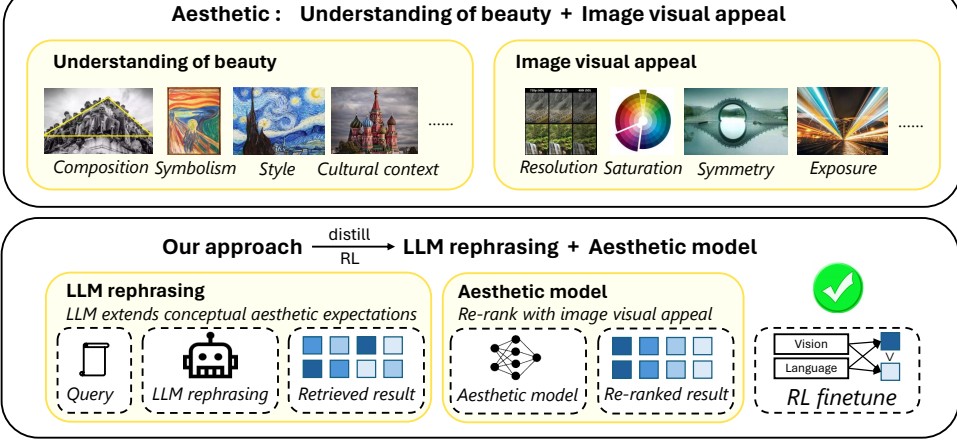

Figure 2: The concept of aesthetic, which inspires our pipeline of alignment. The specific and technical details are shown in Fig. 4.

how to prompt those large multi-modality models toward better aesthetic judgement. (2) We present that LLMs rephrasing of queries can significantly improve the aesthetic scores. (3) We propose a preference-based reinforcement learning method to align existing vision models with human aesthetics. Last but not least, aesthetics is one of the most subjective components of human preferences, and hence we believe our method can be easily generalized to other aspects of human preference.

## 2 Method for Aesthetic Alignment

### 2.1 Model Pretraining

We use both self-pretrained model and open source models [38, 8] for alignment fine-tuning. We pretrain our vision-language model using the adapted CLIP contrastive loss [38], which can be formulated as follows:

$$\mathcal{L}_{\text{text}} = -\sum_{i=1}^{N} \sum_{j=1}^{N} l_i' \log \left( \frac{\exp(s_{ij}/\tau)}{\sum_{k=1}^{N} \exp(s_{ik}/\tau)} \right), \tag{1}$$

$$\mathcal{L}_{\text{image}} = -\sum_{j=1}^{N} \sum_{i=1}^{N} l_j' \log \left( \frac{\exp(s_{ij}/\tau)}{\sum_{k=1}^{N} \exp(s_{kj}/\tau)} \right), \tag{2}$$

where $s_{ij} = \hat{\mathbf{a}}_i^\top \hat{\mathbf{a}}_j$ is the cosine similarity between the embeddings $\hat{\mathbf{a}}_i, \hat{\mathbf{a}}_j$ of the corresponding image and text. $\tau$ is the temperature parameter and $l_i' = (1 - \epsilon) \cdot l_i + \epsilon/N$ indicates a smoothed version of label $l_i$ with a factor of $\epsilon$. The final pretraining loss $\mathcal{L}_{pt}$ is the sum of the text and image losses:

$$\mathcal{L}_{pt} = \mathcal{L}_{\text{text}} + \mathcal{L}_{\text{image}}. \tag{3}$$

The vision model and language model are initialized from the Swin-V2-L [28] and Roberta-L [27], respectively. We leverage several advanced techniques [57, 9, 55] and we leave our detailed insights into the pretraining process and data composition in Appendix. A.

### 2.2 Aesthetically Query Rephrasing with LLMs

According to the explanation of "aesthetic", a query with explicit understanding of aesthetic will potentially benefit the quality of retrieved images. While a typical user's text query can be quite plain, we expect to leverage LLMs (e.g., GPT-3.5-turbo) to enrich such concepts and contents. The participation of LLMs or LMMs is crucial because high level of aesthetic understanding requires their strong reasoning ability. Existing aesthetic models do well to differentiate high and low quality images when they have a great quality gap, but when the gap is small, LMMs surpass all of them significantly via reasoning (Table 13 and 10 in Appendix). This indicates that to make further development of aesthetic understanding, reasoning is essential. While directly labeling a training dataset by LMMs is not acceptable in both latency and cost, using LLMs to reason and extend the query becomes a nice

Figure 3: Effect of LLM rephrasing. All images are retrieved from the same fixed engine. The advancement of LLM rephrasing has clearly enhanced the aesthetic quality of outputs, particularly in expressing abstract notions and stylistic elements.

substitute. In addition, LLMs can further refine the query with the following advantages: (1) Enrich queries with visual details, yielding more aesthetically appealing and expectation-aligned results. (2) Mitigate issues stemming from user search text style, misspellings, and incorrect emphasis.

The impact of the above enhancements will be quantified in Sec. 4.2. Our utilized prompt template can be found in Appendix. F.1, in which a 'method' indicating the rules that query should obey is required. We evaluate four distinct method prompts in Sec. 4.2, and finally advocate the one termed as <k list>:

```
Generate a comma-separated list of succinct object descriptions,
visual details, or stylistic elements to extend the aesthetic understanding
and expectations, ordered from the most to the least significant.
```

An example of query rephrasing is shown in Fig. 3. When we think about "Instagram style", we usually have an imagination of light scene with clean and a little minimalist design. LLM rephrasing adds these elements directly to the query, resulting in a more satisfying retrieval results. In addition to aligning with user's implicit imagination, when the standard of beauty is associated with context of cultural or knowledge, LLM rephrasing can also significantly boost the results. More cases are presented in Appendix. C.

## 2.3 Aesthetic Alignment Fine-tuning

We aim to directly align the retrieval model with human aesthetics, eliminating the multi-stage retrieval system with re-ranker. To obtain the training data for fine-tuning, we leverage public aesthetic models to build a two-stage retrieval system, generating sorted high-quality image sequences. Particularly, given the images retrieved by pretrained model, we utilize well-known semantic (*e.g.*, CLIP [38]) and aesthetic (*e.g.*, CLIPIQA [50], IAP [42] and MANIQA [62]) models as the re-ranker. Although these models are not exactly in the same category (e.g., MANIQA is a no-reference image quality assessment model instead of image aesthetic quality assessment model strictly speaking), we choose these models because during our engineering testing, they perform well as second stage re-rankers. Note that in real world engineering, we can adapt this pipeline to multi-stage system that may further leverage information like click rate, making the pipeline become RLHF.

**Data preprocessing.** We collect our training data using a four-step process:

- We analyze the topic distribution of user queries and employ GPT-3.5-turbo to synthesize $N = 80,000$ pseudo queries that mirrored the distribution of authentic user queries. This procedure can protect user privacy well.

- Each generated query is subjected to a rephrasing process as described in Sec. 2.2. The modified outputs are regarded as rephrased queries.

- For each query, we utilize our pretrained model along with an Approximate Nearest Neighbor (ANN) search algorithm [4] to quickly retrieve the top $K = 400$ images from a 75 million subset of DataComp, using the rephrased query.
- We compute the score of re-ranker (semantic and aesthetic models) for each image in search results.

The final training dataset $\mathcal{D}$ is structured as follows:

$$\mathcal{D} = \{(q_i, \hat{q}_i, \mathbf{T}_i) | i = 1, \ldots, N; \mathbf{T}_i = [\mathbf{x}_i^{(1)}, \mathbf{x}_i^{(2)}, \ldots, \mathbf{x}_i^{(K)}]\}, \qquad (4)$$

where $q_i$ and $\hat{q}_i$ are pseudo query and rephrased pseudo query, and each $\mathbf{x}_i^{(j)}$ is defined as a tuple that contains image $y^{(j)}$ and the re-ranking scores:

$$\mathbf{x}_i^{(j)} = (y^{(j)}, S_{\text{re-rank}}^{(j)}). \qquad (5)$$

**Fine-tuning from AI feedback.** We model the retrieval problem as a reinforcement learning problem: for a given search query $q$ and an image database $\mathcal{Y} = \{y_n\}$, we denote the retrieval system with learnable parameters as the policy model $\pi_\theta(y|q; \mathcal{Y})$. For some of the retrieved images, e.g., $y_i$ and $y_j$, we can establish a preference $y_i > y_j$ (or $y_i < y_j$) to signify that image $y_i$ (or $y_j$) is a preferred retrieval response. Assume that these preferences are generated by some underlying reward model $r_\phi(y, q)$, e.g., human/AI scorer, and aesthetic model. Reinforcement learning maximizes the expectation of rewards while using KL divergence for regularization to prevent training collapse and overfitting:

$$\max_{\pi_\theta} \mathbb{E}_{q \sim \mathcal{D}, y \sim \pi_\theta(y|q; \mathcal{Y})}[r_\phi(y, q)] - \beta \mathbb{D}_{KL}[\pi_\theta(y|q; \mathcal{Y}) \| \pi_{ref}(y|q; \mathcal{Y})]. \qquad (6)$$

Here, $\pi_{ref}$ is the reference model (i.e., the pretrained model). Following DPO [39], by choosing Bradley-Terry model [3] to formulate preference distribution, we can use the following policy objective loss to maximize the reward:

$$\mathcal{L}_{dpo} = -\mathbb{E}_{(q, y_w, y_l) \sim \mathcal{D}_{po}} \left[ \log \sigma \left( \beta \log \frac{\pi_\theta(y_w|q; \mathcal{Y})}{\pi_{ref}(y_w|q; \mathcal{Y})} - \beta \log \frac{\pi_\theta(y_l|q; \mathcal{Y})}{\pi_{ref}(y_l|q; \mathcal{Y})} \right) \right], \qquad (7)$$

where $y_w$ is the preferred sample compared to $y_l$. In retrieval scenario, given a user search query $q$, we build the partially ordered dataset $\mathcal{D}_{po}$ for training by establishing those ordered pairs: $\mathcal{D}_{po} = \{(q, y_i, y_j) | y_i < y_j\}$. The probability that multimodal based policy model return response $y_i$ is given by the normalized cosine similarity:

$$\pi_\theta(y_i|q; \mathcal{Y}) = \frac{\cos(f_v^\theta(y_i), f_l^\theta(q))}{\sum_{y_k \in \mathcal{Y}} \cos(f_v^\theta(y_k), f_l^\theta(q))}. \qquad (8)$$

Here, $f_v^\theta$ and $f_l^\theta$ represent the vision and language encoders of the multimodal model, respectively. It is easy to observe that the denominator part of $\pi_\theta(y|q; \mathcal{Y})$ will be cancelled out in $\mathcal{L}_{dpo}$, thus in actual calculations, we only need to calculate the cosine similarity of the corresponding images and queries, which also makes $\mathcal{L}_{dpo}$ independent of the image database $\mathcal{Y}$. Compared to DPO [39], we utilize an ordered sequence to obtain samples for adapting the DPO loss, allowing producing $O(n^2)$ preference pairs from a sequence of length $n$. This approach significantly enhances the data utilization rate, making the modified DPO algorithm more scalable.

Following InstructGPT [36], we also integrate the pre-training loss $\mathcal{L}_{pt}$ to stabilize the training process and maintain retrieval capability. Consequently, the composite loss function formulated for fine-tuning is expressed as:

$$\mathcal{L} = \mathcal{L}_{dpo} + w_{pt}\mathcal{L}_{pt}. \qquad (9)$$

**Construction of $\mathcal{D}_{po}$.** We illustrate the construction of the partially ordered dataset $\mathcal{D}_{po}$ in Fig. 4. For each query, images are intermittently selected from the retrieved results of the rephrased query at intervals defined as stride, aiming to amplify the quality discrepancy among the chosen images. Subsequently, these images are arranged into a matrix with $u$ rows and $v$ columns, following a row-major format. Samples within each row are sorted according to the score of the re-ranker. The re-ranker, which is finally designed by assembling open-source models, is evaluated in Appendix. E. We then define the column dimension as the aesthetic dimension, since the samples in each row are sorted aesthetically. The row dimension is defined as the semantic dimension because semantic relevance varies across different rows. We extract all rows and columns to obtain $u + v$ ordered sequences with the form $y_1 > y_2 > \ldots > y_k$, resulting in $C_k^2$ pairs of $(y_i, y_j)$ in each sequence. To this end, a total of $uC_v^2 + vC_u^2$ partial order pairs can be produced for each query. Note that numerous operations in this process can be tensorized and executed in parallel, thus the time cost is low.

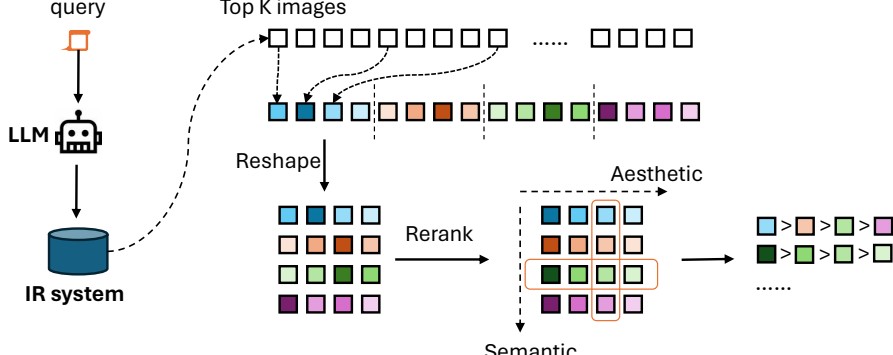

Figure 4: An example illustration for the construction of partially ordered pair dataset.

# 3 Benchmarking Human Preference

Standard retrieval benchmarks, including MSCOCO [25] and Flickr30k [63], lack the aesthetic evaluation. Aesthetic datasets, *e.g.* AVA [32], can only be used to evaluate the accuracy of an aesthetic prediction model, which needs additional efforts for retrieval system evaluation. Therefore, we introduce the following two novel benchmarks to assess whether a retrieval model can align with human aesthetics well: (1) testing model preference by human-labeled dataset, and (2) using GPT-4V [35] to determine the relative performance of two systems.

## 3.1 Human Preference of Image Retrieval (HPIR)

We introduce HPIR, a test set of human preference alignment. It leverages 150 pseudo queries for testing, which are generated using LLMs by requesting an aligned distribution with the user's topics. For each query, we combine the results of multiple search engines and obtain 10 images, which are divided into two groups (A and B). Human labelers are asked to evaluate the two groups of images, determining which group is more precise and visually appealing. To ensure robustness, each comparison is annotated for 30 times. We also scrutinize the annotation time and order consistency (Sec. 3.2) to guarantee the quality. The label that predominated in the 30 annotations is designated as the golden label. Let $N_{pos}$ be denoted as the number of labelers that assign the golden label, and $N_{neg}$ as the remaining number. We define the confidence score $w_c$ (exemplified in Fig. 14 of Appendix) of this annotation as:

$$w_c = \frac{2N_{pos}}{N_{pos} + N_{neg}} - 1 \in [0, 1]. \tag{10}$$

This confidence level has a similar effect to variance, and the variance formula for human labelers in aesthetic annotations can be easily calculated as follows:

$$var = \frac{2N_{pos}N_{neg}}{(N_{pos} + N_{neg})^2}. \tag{11}$$

To evaluate a model/search engine, we task it with discerning the better group between A and B for all queries, based on a designated criterion. Then the HPIR metric $M_{asp}$ ($asp$ stands for either accuracy or aesthetics.) is assessed by comparing the selections of the model/engine to the human-annotated golden labels. $M_{asp}$ is formulated as a confidence-weighted average over the queries:

$$M_{asp} = \frac{\sum_{query} w_c \cdot \mathbb{1}\{choice = golden\_label\}}{\sum_{query} w_c}, \tag{12}$$

where $\mathbb{1}\{choice = golden\_label\}$ is an indicator that equals 1 when the model's choice matches the golden label, and 0 otherwise. This method can effectively assess the degree of alignment between a model and human preferences. For instance, to evaluate the CLIP [38], we simply compute the average CLIP similarities (to query) on group A and B, choosing the group with the higher average as the model's choice. More details about data distribution, baseline results and aesthetic model evaluation can be found in Sec. 4.1 and in Appendix. E.

## 3.2 GPT-4V Win Rate

LMMs have shown their strong abilities across numerous tasks. Thus, we directly compare two retrieval models/systems using GPT-4V [35]. Emulating the AlpacaEval [24] approach from LLMs, we first conduct image searches for a collection of queries using two retrieval systems, R1 and R2. We then concatenate results in R1 and R2 into one large image and employ GPT-4V as the judge to assess which system performs better. We note that GPT-4V tends to prefer the first row when the results from both systems are comparable, a tendency that mirrors human behavior. To address the bias, we introduce an **order-consistency** (OC) strategy, where we initially place images from R1 on the first row and images from R2 on the second for evaluation, then invert their positions for a separate assessment. A visualization and more detailed description is provided in Appendix. D.1. If two assessments have conflicting conclusions, we say the two results are similar. Lets denote the number of R1 wins as $N_w$, loses as $N_l$ and similar as $N_s$. System R2 serves as the baseline. We define the win rate of R1 as $R_{win}$, and a win-and-similar rate as $R_{win\&similar}$:

$$R_{win} = \frac{N_w}{N_w + N_l}, \tag{13}$$

$$R_{win\&similar} = \frac{N_w + N_s}{N_w + N_s + N_l} = 1 - \frac{N_l}{N_w + N_s + N_l}. \tag{14}$$

Unlike HPIR, this approach lacks the supervision of human labelers, necessitating meticulous design and validation to ensure its soundness. We thus leverage HPIR feedback to filter various prompts and evaluation methods, ultimately selecting a prompt format referred to as <ranker> (see Appendix. G). Detailed experiments, shown in Appendix. D.2, yield that with this prompt and order-consistency, GPT-4V can present comparable aesthetic judgments to humans.

## 4 Experiments

In this section, we present our main experiments. More results, including ablations, benchmark evaluations (HPIR and GPT-4V judge), and LLM rephrasing, are in Appendix. B - F.

### 4.1 Details and Evaluations of Alignment Fine-tuning

In the alignment fine-tuning loss, the $\mathcal{L}_{pt}$ component is configured identically to the pretraining phase described in Sec. 2.1, encompassing batch size, temperature, and data, with a weight of $w_{pt} = 1.0$. For the remaining components, each batch comprises 128 queries. The overall learning rate is fixed to $lr = 5 \times 10^{-5}$. The partially ordered set $\mathcal{D}_{po}$, as discussed in Sec. 2.3, is derived using $u = v = 5$, and a stride of 10.

We conduct the experiments with two other state-of-the-art models: CLIP [38] and DataComp [8]. The image encoders of CLIP and DataComp are ViT-L/14 models, trained on a private 400M dataset and on the DataComp-1B dataset, respectively. We report their performance on classic retrieval benchmarks (ImageNet1K [6] zero-shot classification, MSCOCO [25] T2I retrieval Recall@1, and Flickr30K [63] T2I retrieval Recall@1) and on the proposed HPIR in Table 1. It is not surprising that our model performs worse to DataComp on MSCOCO and Flickr30K, since our training budget is much smaller than DataComp. We can further observe that our alignment fine-tuning does not significantly impact the retrieval capability, but it greatly enhances the aesthetic scores of the retrieval results, surpassing both original CLIP and DataComp. While fine-tuning on the CLIP and DataComp, a similar increase is observed in aesthetic ability, demonstrating the generalization of our method.

Table 1: Performance on traditional retrieval benchmarks and our proposed aesthetic alignment dataset. PT indicates pre-training, and RLFT indicates our alignment fine-tuning.

| Model | Retrieval metrics (%) | | | HPIR (%) | |
|---|---|---|---|---|---|
| | ImageNet1K-ZS | MSCOCO | Flickr30K | Accuracy | Aesthetic |
| CLIP[38] | 76.2 | 37.8 | 68.7 | 68.1 | 62.1 |
| DataComp[8] | 79.2 | 45.7 | 73.4 | 71.8 | 62.7 |
| Ours-PT | 82.1 | 40.2 | 66.5 | 66.2 | 59.4 |
| CLIP + RLFT | 79.4 | 38.1 | 67.2 | 73.1 | **71.7** |
| DataComp + RLFT | 81.7 | 45.1 | 72.2 | 74.4 | **71.9** |
| Ours-RLFT | 82.2 | 40.2 | 66.4 | 71.7 | **67.6** |

In Table 2, we report the system-level comparison results with other models, approaches and two commercial search engines (Bing image search [31] and Getty search [13]). We report the win-

and-similar rate here because it is more in line with the user's thinking when choosing products (see other indices and details in Appendix. D.3). Experiments are conducted on a database of 15M images extracted from DataComp-1B and our internal database with 8M images. The experiments demonstrate the effectiveness of our alignment fine-tuning, the superiority of our final model, and the gap with commercial web-scale products. Our finetuned model shows comparable performance with 2-stage approach, yielding a possible latency and pipeline optimization to real world products (see win rate in Appendix. D.3). Systematically speaking, despite the vast difference in the size of the databases, our model still achieved a win-and-similar rate of 50%∼60% in comparisons with Bing image search and Getty search. Additional human user evaluations of selected experiments also validate the reliability of the GPT-4V judge. These labeling processes can be seen as real world A/B test user studies.

Table 2: System level comparison results. Mark † indicates using LLM rephrasing. DataComp-15M is a 15 million subset of DataComp-1B dataset [8].

| ID | System A | | | System B | | | A to B win & similar rate (%) | |
|---|---|---|---|---|---|---|---|---|
| | Name | Database | Stages | Name | Database | Stages | Accuracy | Aesthetic |
| 1 | Ours-FT | Datacomp-15M | 1 | Ours-PT | Datacomp-15M | 1 | 72.0 | 78.7 |
| 2 | Ours-FT | Datacomp-15M | 1 | CLIP | Datacomp-15M | 1 | 74.0 | 77.3 |
| 3 | Ours-FT | Datacomp-15M | 1 | DataComp | Datacomp-15M | 1 | 69.3 | 67.3 |
| 4 | Ours-FT | internal-8M | 1 | Ours-PT + Rerank | internal-8M | 2 | 68.0 | 70.0 |
| 5 | Ours-FT † | internal-8M | 1 | Bing search | web | >2 | 45.3 | 57.3 |
| 6 | Ours-FT † | internal-8M | 1 | Getty search | Getty Images | >2 | 62.7 | 62.7 |
| | Human labeler judger (User study) | | | | | | | |
| 1' | Ours-FT | Datacomp-15M | 1 | Ours-PT | Datacomp-15M | 1 | 65.3 | 74.7 |
| 4' | Ours-FT | internal-8M | 1 | Ours-PT + Rerank | internal-8M | 2 | 66.9 | 71.4 |
| 5' | Ours-FT † | internal-8M | 1 | Bing search | web | >2 | 49.1 | 56.6 |
| 6' | Ours-FT † | internal-8M | 1 | Getty search | Getty Images | >2 | 63.3 | 61.3 |

## 4.2 Effect of LLM Rephrasing

Here, we test different method prompts, which correspond to {method} as outlined in the template in Sec. 2.2, for search query rephrasing. We task GPT-3.5-turbo with the job of rephrasing queries to an approximate word count of 50. The method <k list>, as introduced in Sec. 2.2, enumerates additional descriptions for the query. The <detail> method encourages the model to elaborate with more specifics, while <kw dict> instructs the LLM to enumerate keywords followed by detailed expansions for each. We employ two verification metrics as described in this paper: HPIR and GPT-4V win rate, with the latter benchmarked against the original query's baseline results. The detailed prompts are described in the Appendix. F.2.

Table 3 reports the evaluation results, including the average scores from aesthetic models. The empirical evidence from all three metrics suggests that the utilization of LLMs for query rephrasing has enhanced the aesthetic appeal of the search results. We choose win rate to report because it brings clearer clues of which prompt is better (see other indices and details in Appendix. D.4).

Table 3: Evaluation of different method prompts for LLM rephrasing on HPIR and GPT-4V win rate. Scores from aesthetic models are also provided.

| prompt | Aesthetic Scores (5 Avg.) | | | HPIR (%) | | GPT-4V win rate (%) | |
|---|---|---|---|---|---|---|---|
| | CLIPIQA | IAP | MANIQA | Accuracy | Aesthetic | Accuracy | Aesthetic |
| original | 0.8544 | 4.8047 | 0.4296 | 66.19 | 59.36 | - | - |
| <detail> | 0.8787 | 5.0698 | 0.4279 | 65.25 | 68.24 | 63.51 | 68.75 |
| <k list> | 0.8768 | 5.0772 | 0.4384 | 68.95 | 72.42 | 53.37 | 67.86 |
| <kw dict> | 0.8678 | 5.0140 | 0.4345 | 69.17 | 69.93 | 53.57 | 63.83 |
| repeat | 0.8554 | 4.9525 | 0.4395 | 67.96 | 65.67 | 62.00 | 63.16 |
| <reorg> | 0.8742 | 4.9925 | 0.4463 | 68.33 | 67.85 | 59.65 | 64.44 |

To investigate the mechanisms by which query rephrasing yields enhancements, we further evaluate two additional rephrasing methods for extending the length of the original query: the "repeat" method, which involves duplicating the original query $n$ times, and the <reorg> method, which entails prompting the LLM to reformulate the query in diverse linguistic styles, then repeating it $n$ times

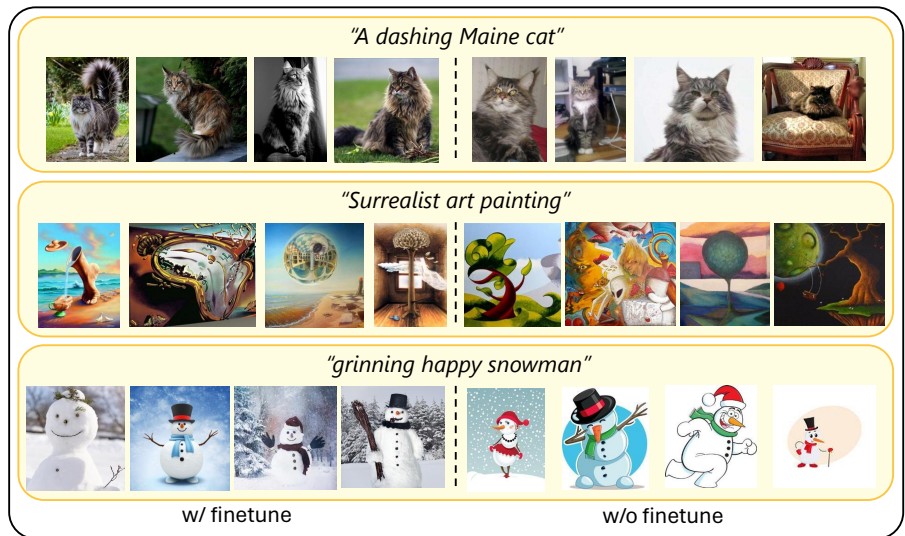

*"A dashing Maine cat"*

*"Surrealist art painting"*

*"grinning happy snowman"*

w/ finetune        w/o finetune

Figure 5: Qualitative comparison of top-4 retrieval results between models with and without our proposed alignment fine-tuning.

without incorporating additional details. As shown in Table 3, simply enlarging the length of the query, even in the absence of new details, can enhance the aesthetic performance. Leveraging LLMs to deepen the comprehension of the query and enrich the visual specifics allows for further aesthetic improvement in retrieval tasks. We further summarize 2 possible reasons for this phenomenon in Appendix. F.3.

## 5 Cases Study and Qualitative Comparison

Fig. 5 shows the qualitative comparison between our fine-tuned model and pretrained model, where we retrieve top-4 images from the 75M subset of DataComp. It can be observed that the alignment fine-tuning endows the model with the capability to retrieve images with vivid background, rich texture details, and dynamic color contrast, leading to more aesthetically pleasing search engine.

More comparison result and analysis with and without LLM rephrasing using our fine-tuned model can be found in Appendix. C. With LLM rephrasing, the retrieved images exhibit remarkable improvement on visual coherence and enriched details. The styles of the images become more consistent with the search intent, capturing samples that align closely with human expectation.

## 6 Related Work

**Vision Language Models.** The availability of web-scale image-text pairs has sparked the research of vision-language models [7, 21, 23, 29, 46, 51, 67]. The pioneering works, CLIP [38] and ALIGN [14], utilize contrastive loss during training to achieve remarkable generalization capabilities. Subsequent works [61, 65, 44] have expanded image-text contrastive to a wider scope. BeiT3 [52], CoCa [64] and BLIP [20] further explore other pretraining methods. More recently, several large multi-modal models have emerged [26, 53, 12]. While most methods have shown strong retrieval capabilities, they often overlook aesthetics, frequently retrieving results with poor visual quality. Our work aims to fill this gap, focusing on designing a vision-language model with aligned aesthetics with humans.

**Reinforcement Learning from Human Feedback (RLHF).** RLHF has been widely adopted in LLMs [36, 45, 68, 49, 39]. Typically, a reward model is trained as a proxy for human preferences, providing feedback for model tuning. Recently, researchers focus and apply RLHF technique to computer vision [37, 16, 41]. In image generation, Lee *et al.* [18] and ImageReward [60] utilize the reward modeling for text-to-image tasks. To the best of our knowledge, our work is the first to focus on aligning human intents with the fundamental task of text-image retrieval.

**Image Aesthetics Assessment.** Numerous aesthetic evaluation protocols have been proposed [17, 47, 11, 22, 34, 42]. Kong *et al.* [17] propose relative ranking to model photo aesthetics. NIMA [47] assigns a score distribution to each photo, which captures the subjective variations in human aesthetic preferences. MANIQA [62] considers the no-reference image quality assessment task and employs a multi-scale attention mechanism to analyze images across various scales and regions. CLIPIQA [50] tries to understand image content and trains the model by comparing the quality of images. In this work, we adopt a weighted combination of existing models [62, 50, 42, 38] to provide supervision for our training. In addition, several IAA datasets (AVA [32], PN [15], and CHUNK-PQ [30]) have been proposed for evaluating aesthetic models. However, in retrieval setting, we need to compare the aesthetic levels of two image groups corresponding to a shared query. Existing datasets cannot satisfy the needs in retrieval scenario.

## 7 Potential Social Impacts and Risks

We state that all labelers (mostly university students) were informed of all the uses of their labels, and they agreed on the use. All experiments were conducted with open-source datasets and proprietary data for which we own the full copyright.

Our work may cause potential social impacts and risks:

- Diversity and quality trade-off: The LLM rephrasing stage may increase the quality while decrease the diversity, which generally does more good than harm, but in cases where more diversity is required, several simple adjustments can be made to adapt. For example, repeating the rephrasing may benefit both direct inference and training models. In addition, increasing the pre-training loss weight during fine-tuning may also be an option to balance the trade-off.

- Potential biases: LLMs are trained with massive internet-scale data and may therefore contain potential biases. This may reinforce the biases of the retrieval system. For example, when searching for the word 'nurse', the vector search system using the CLIP feature returns approximately 90% female images and 10% male images. However, when this system uses LLM rephrasing as preprocessing, it results in nearly 100% female images. This is because the rephrased sentence describes a female nurse. However, our approach also provides an opportunity to supervise and eliminate some biases. A vector search system using the CLIP feature is a black box, and we cannot predict or control when it may retrieve harmful content. By using LLM rephrasing, we can trace intermediate information and debug the system. We can further improve LLMs or implement a text-based filter.

- Specific RAI problems: As we mentioned, our algorithm can be adapted to other areas of RAI problems, such as nationality and race. While our approach may benefit many general cases, it may potentially ignore minorities, as aesthetics and ethics may differ across various cultures and regions. Specific adaptations should be made to both LLMs and vision models in accordance with the 'no free lunch' theorem.

## 8 Conclusion

In this paper, we attempted to align image retrieval models with human aesthetics. We presented a preference-based reinforcement learning method to align retrieval models with human aesthetics by distilling knowledge from LLMs reasoning and aesthetic models. Extensive experiments demonstrated the effectiveness of our method, showing a possible alternative to the multi-stage retrieval pipeline. Finally, we discussed the potential social impacts and risks. We believe our proposed approach can become a general practice for other misalignment problems in computer vision.

**Acknowledgement** This project was supported by the Fundamental Research Funds for the Central Universities (2242024k30035).

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

# A  Pre-training Details

We pretrain our vision-language model, whose vision model (i.e., swin transformer) has been pretrained via SimMIM [57] and subsequently fine-tuned on ImageNet-22k [6] prior to the CLIP pretraining. To accommodate a larger batch size and an increased number of negative samples with limited resources, we implemented gradient accumulation and a queue dictionary, similar to MoCo [9]. In line with iCLIP [55], we incorporated ImageNet-22k to bolster the model's capability for short query and keyword searches. Thus, comparison of Table 1 on ImageNet is not totally fair, we report it only for a representative of keyword search capability.

Table 4 lists the composition and amount of the data during model pretraining. Throughout the training process, we employ a fixed learning rate $lr = 5 \times 10^{-4}$. Leveraging gradient accumulation and a queue, we enable a batch size of $32k$ for contrastive learning. Our training method, similar to that of CLIP, incorporates the NCE Loss complemented by a label smoothing factor of $0.1$. Difference from CLIP, we initialize the temperature parameter to 0.05 and treat it as a learnable parameter. The computational resources include 256 NVIDIA V100 GPUs.

Table 4: The composition and amount of data during pretraining. SCVL indicates our self-collected image-text pair dataset.

| Dataset | Samples (M) | Used Samples (M) | Used Epoches |
|---|---|---|---|
| DataComp [8] | 1000 | 5000 | 5.0 |
| SCVL | 200 | 2000 | 10.0 |
| ImageNet22K [6] | 14 | 140 | 10.0 |
| Total | 1214 | 7140 | - |

Table 5 shows the specific hyper-parameters for pre-training, including the data augmentation and model settings. The loss curve and gradient norm curve during the training process are shown in Fig. 6, where the deep blue line is the curve after smoothing. Fig. 7 demonstrates the curve of the retrieval evaluation metrics during the pre-training process, on three datasets (ImageNet1k-ZS, MSCOCO, Flickr30k).

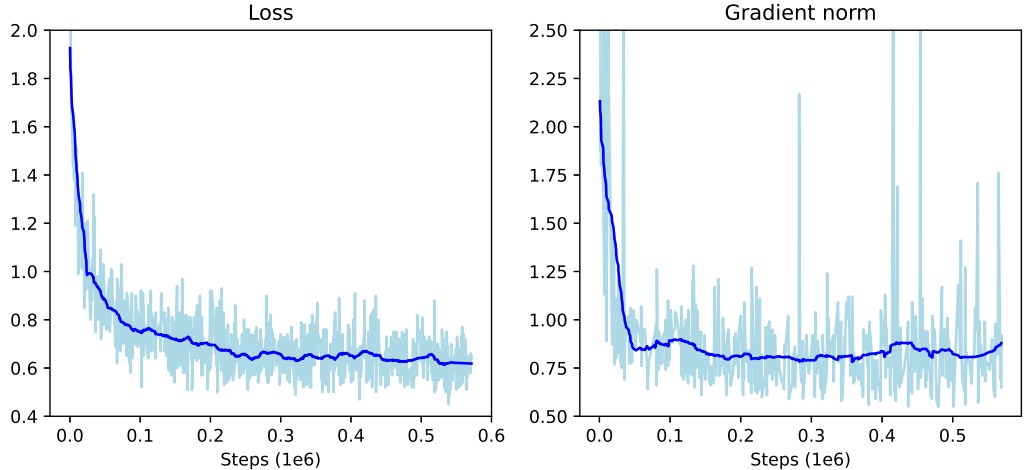

Figure 6: Loss and gradient norm curves during pre-training.

Table 5: Details of pre-training hyper-parameters.

| Training | | Loss | |
|---|---|---|---|
| learning rate | 5e-04 | loss | NCE |
| total batch size | 32768 | label smoothing | 0.1 |
| batch size per GPU | 32 | **Data Augmentation** | |
| lr dacay - v | 0.9 | auto aug | rand-m9-mstd0.5-inc1 |
| lr dacay - l | 0.9 | color jitter | 0.4 |
| lr scheduler | Fixed | cutmix | 0.0 |
| warm up steps | 5000 | mixup | 0.0 |
| weight decay | 0.05 | **Model** | |
| optimizer | AdamW | tau init | 0.05 |
| dropout | 0.0 | tau learnable | True |
| drop path | 0.0 | project size | 1024 |
| grad clip | 3.0 | average pooling | True |
| amp | True | vmodel | Swin-Large |
| opt level | O1 | lmodel | RoBERTa-Large |

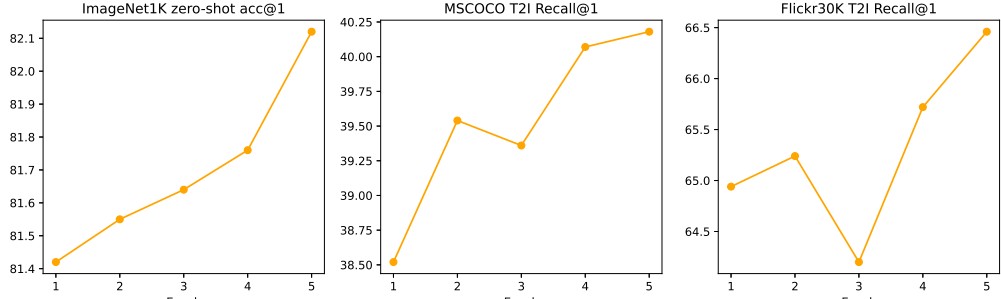

Figure 7: Evaluation curves on retrieval benchmarks during pre-training.

# B Alignment Fine-tuning Details and Ablations

## B.1 Fine-tuning Details and Curves

We show the details of fine-tuning hyper-parameters in Table 6. The whole training takes about 650 steps on 4 NVIDIA A100 GPUs. According to the ablation experiments in the following sections, we found that using strong data augmentations have positive influence on HPIR-Accuracy and retrieval metrics (mostly focus on semantic accuracy), but they may harm aesthetic performance. Thus we do not add any data augmentation during alignment fine-tuning. Fig. 8 shows the loss curves and gradient norm curve during fine-tuning. Note that the total loss is the weighted sum of DPO loss and pre-training loss. Fig. 9 shows the performance on retrieval benchmarks and HPIR during fine-tuning. It can be observed that the retrieval performance stays stable. While some fluctuation happened on HPIR curves, we believe this is because of the unavoidable subjective nature of HPIR, as both options (groups) of the dataset have close appearances.

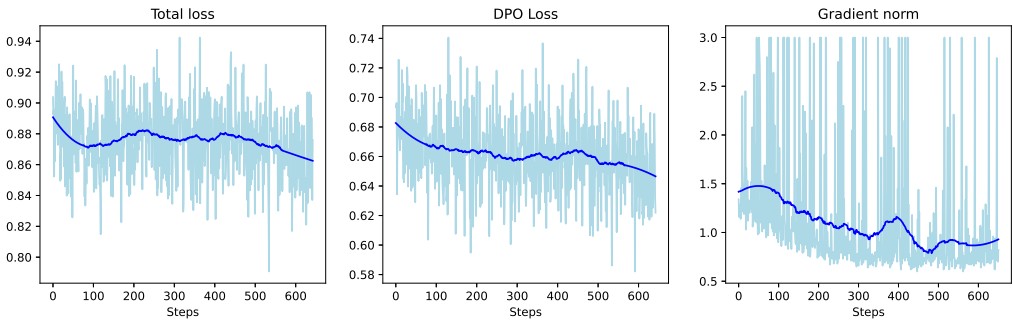

Figure 8: Total loss, DPO loss and gradient norm curves during fine-tuning.

Table 6: Details of fine-tuning hyper-parameters.

| Training | | Loss | |
|---|---|---|---|
| learning rate | 5e-5 | pre-train loss | NCE |
| total batch size | 128 | label smoothing | 0.1 |
| batch size per GPU | 32 | $w_{pt}$ | 1.0 |
| lr dacay - v | 0.9 | finetune loss | Ranked DPO |
| lr dacay - l | 0.9 | $\beta$ | 0.05 |
| lr scheduler | Cosine | $u$ | 5 |
| warm up steps | 200 | $v$ | 5 |
| weight decay | 0.0 | stride | 10 |
| optimizer | AdamW | **Data augmentation** | |
| dropout | 0.0 | auto augmentation | None |
| drop path | 0.0 | color jitter | 0.0 |
| grad clip | 3.0 | cutmix | 0.0 |
| amp | True | hflip | 0.0 |
| opt level | O1 | mixup | 0.0 |

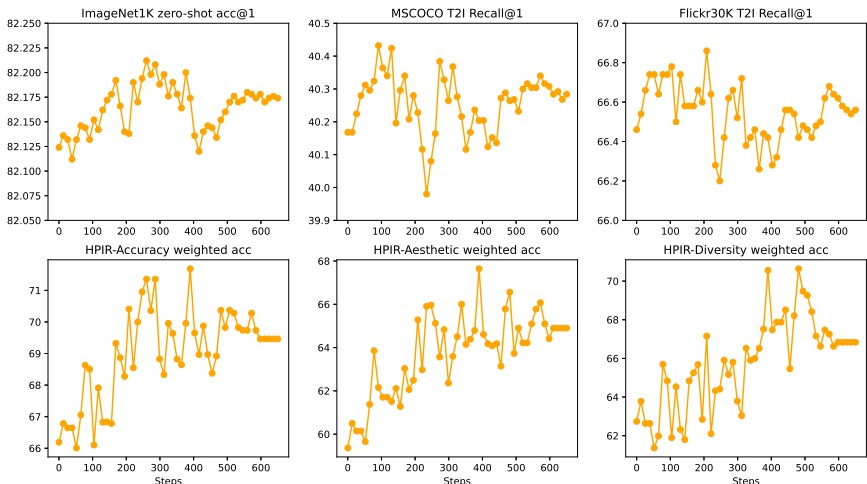

Figure 9: Evaluation curves on benchmarks and HPIR during alignment fine-tuning.

## B.2 Effect of 2-D sampling of $\mathcal{D}_{po}$

In order to perform a solid comparison, all experiments in Table 7 are set to similar budgets and the same hyper-parameters. The number of partial order pairs used for one query, as described in the main paper, is $uC_v^2 + vC_u^2$. Thus we choose the $u$ and $v$ to have similar number of partial order pairs. $|\mathcal{D}_{po}|$ represent the total size of fine-tuning dataset.

We first ablate the impact of the two dimensional sampling strategy in the construction of $\mathcal{D}_{po}$. In Table 7, the first line ($u = 15, v = 1$) represent that we only sample from semantic dimension, and the last line of the first block ($u = 1, v = 15$) shows the performance of sampling only from aesthetic dimension. As expected, the two dimensions bring different inductive biases derived from two aspects, and hence assembling both can benefit the alignment fine-tuning. The larger stride shows better result within a certain range, but few influence when it is large enough.

We also have to re-emphasize that although we named $u$ as the semantic dimension, however, because the ranked sequence is retrieved from rephrased query (instead of the original query), the sequence may contain aesthetic benefits from LLM rephrasing. Thus $u$ dimension can also help aesthetic performance. As a result, modifying the $u, v$ may not cause a remarkable change in results.

Table 7: Experiments on the effect of the construction of partial order set $\mathcal{D}_{po}$. $\mathcal{Q}$ is the number of queries.

| stride | $u$ (semantic) | $v$ (aesthetic) | $|\mathcal{D}_{po}|$ | Accuracy | Aesthetic |
|---|---|---|---|---|---|
| | 15 | 1 | $105 \times |\mathcal{Q}|$ | 72.1 | 66.5 |
| | 8 | 3 | $108 \times |\mathcal{Q}|$ | 71.3 | 67.1 |
| 10 | 5 | 5 | $100 \times |\mathcal{Q}|$ | 71.7 | 67.6 |
| | 3 | 8 | $108 \times |\mathcal{Q}|$ | 70.3 | 65.4 |
| | 1 | 15 | $105 \times |\mathcal{Q}|$ | 68.1 | 67.5 |
| 5 | | | | 68.7 | 64.4 |
| 10 | 5 | 5 | $100 \times |\mathcal{Q}|$ | 71.7 | 67.6 |
| 15 | | | | 70.9 | 67.2 |

## B.3 Loss Selection

In addition to the ranked DPO loss as we introduced, we also adapt other losses to our scenario. We introduce them in the follows:

### B.3.1 RRHF Loss.

RRHF [66] is a more simple approach that do not need the reference model. In our scenario, it is formulated as follow:

$$\mathcal{L}_{rrhf} = -\mathbb{E}_{(q,y_w,y_l)\sim\mathcal{D}_{po}} \left( \pi_\theta(y_w|q;\mathcal{Y}) - \pi_\theta(y_l|q;\mathcal{Y}) \right), \tag{15}$$

$$\mathcal{L} = \mathcal{L}_{rrhf} + w_{pt}\mathcal{L}_{pt}. \tag{16}$$

### B.3.2 IPO Loss.

IPO [1] is also a general objective for RLHF. It is based on maximizing a non-linear function of preferences and keeps KL regularization. Similar to DPO, we adapt IPO to a ranking version, which is formulated as follows:

$$\mathcal{L}_{ipo} = -\mathbb{E}_{(q,y_w,y_l)\sim\mathcal{D}_{po}} \left( \log \frac{\pi_\theta(y_w|q;\mathcal{Y})}{\pi_{ref}(y_w|q;\mathcal{Y})} - \log \frac{\pi_\theta(y_l|q;\mathcal{Y})}{\pi_{ref}(y_l|q;\mathcal{Y})} - \frac{1}{2\beta} \right)^2, \tag{17}$$

$$\mathcal{L} = \mathcal{L}_{ipo} + w_{pt}\mathcal{L}_{pt}. \tag{18}$$

For each loss, we tuned their hyper-parameters and reported their best results in Table 8. Firstly, all losses bring growth to the HPIR results. This proves again that the way we built the training set provides valid, learnable patterns. IPO and DPO perform similar results. RRHF loss, as it does not contain regularization terms, can not scale well. The peak result of RRHF loss is obtained at about 100 steps (of 650 steps in total). Then its retrieval ability (i.e., retrieval benchmark results) decreases rapidly to an unacceptable range.

Table 8: Results of alignment fine-tuning with different loss objective functions.

| Fine-tune Loss | Retrieval benchmarks | | | HPIR | |
|---|---|---|---|---|---|
| | ImageNet1K-ZS | MSCOCO | Flickr30K | Accuracy | Aesthetic |
| None (only pre-training) | 82.1 | 40.2 | 66.5 | 66.2 | 59.4 |
| RRHF | 82.1 | 40.1 | 67.6 | 68.4 | 62.2 |
| Ranked IPO | 81.4 | 39.7 | 65.3 | 70.2 | 66.8 |
| Ranked DPO | 82.2 | 40.2 | 66.4 | 71.7 | 67.6 |

## B.4 Impacts of Data Augmentation

Data augmentation, as it is known to all, bring benefits to a large range of computer vision tasks. Most of those tasks require model generalization capabilities on semantic dimension. Data augmentation greatly brings in important inductive biases, e.g., translation invariance, robustness of dealing occlusion, watermark, color change. However, we found in the aesthetic related tasks, some data augmentations may harm the performances.

In Table 9, the best results are obtained by implementing the same image transform with the evaluation. The evaluation transform only contains an image size transform and color normalization. We separately add different data augmentations to the evaluation transform, and find that they affect aesthetic indicators to varying degrees, while the accuracy aspect of HPIR metric may gain benefits.

Table 9: Ablation of different data augmentations. Different strategies are added to the base transform separately.

| Data augmentation | | HPIR | |
| --- | --- | --- | --- |
| type | values | Accuracy | Aesthetic |
| Eval-transform | - | 71.7 | 67.6 |
| + auto-aug | rand-m9-mstd0.5-inc1 | 74.4 | 66.9 |
| + random erase | 0.25 | 72.5 | 65.2 |
| + color jitter | 0.4 | 71.4 | 64.7 |

## C    More cases study of LLM rephrasing

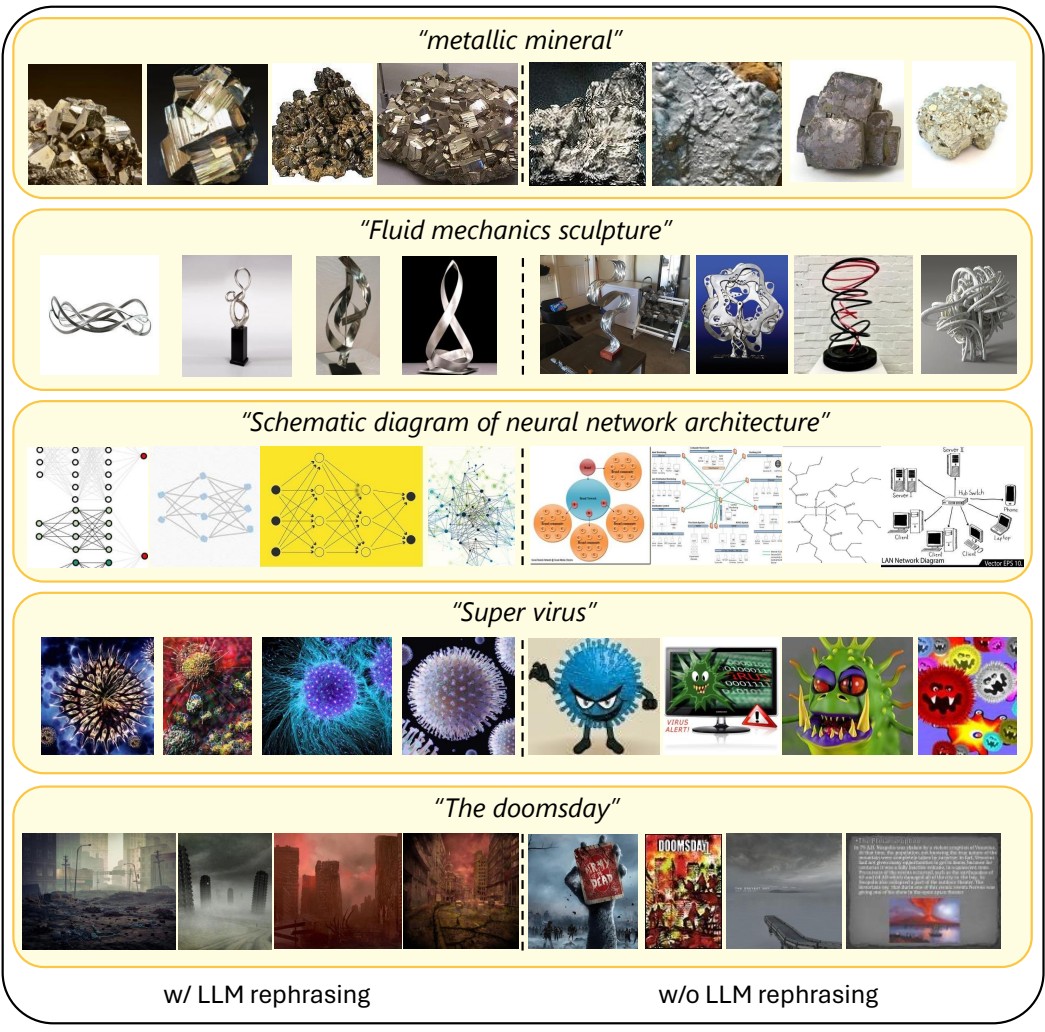

Figure 10: Qualitative comparison of top-4 images with and without LLM rephrasing. When user search query, they may implicitly have a expectation or imagination, LLM rephrasing help extent the imagined elements.

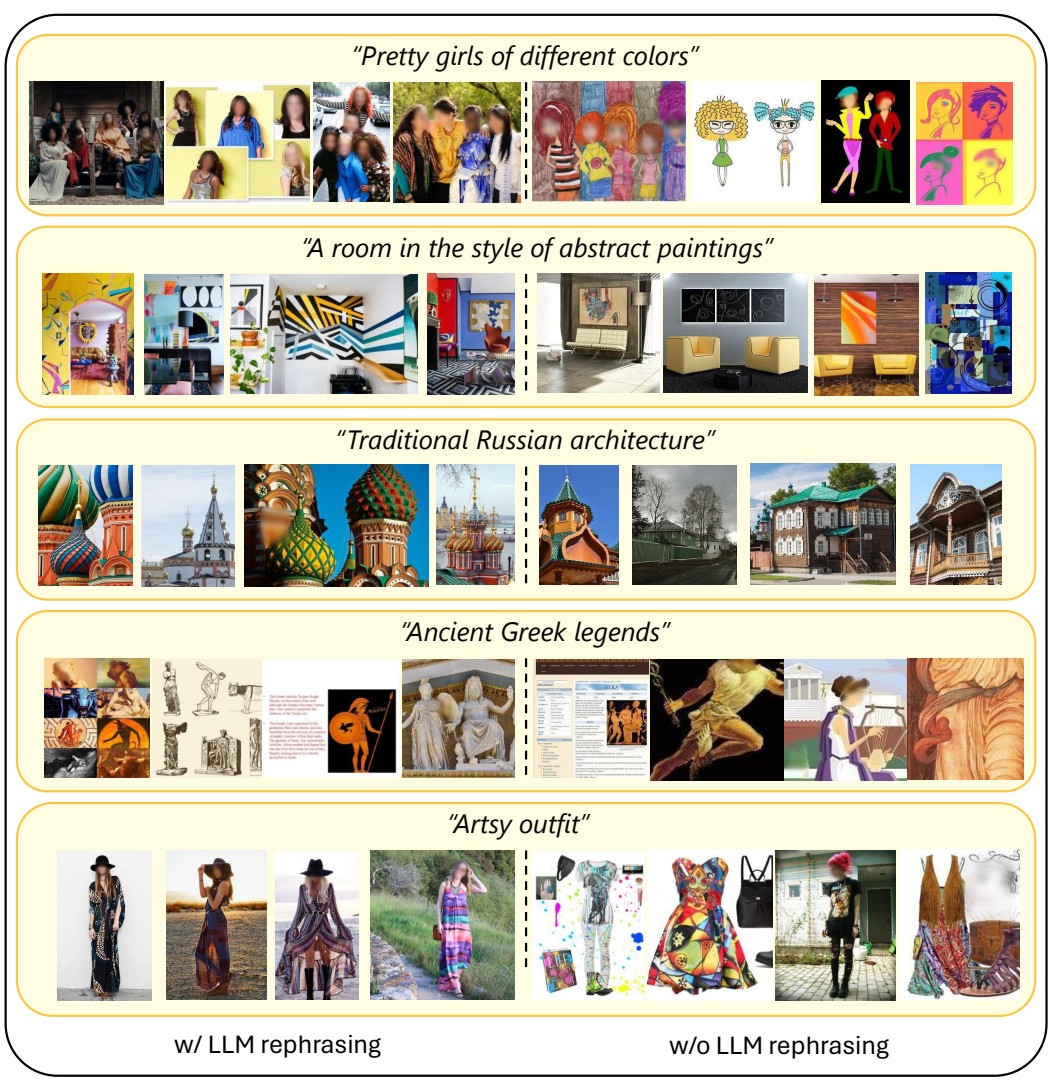

Figure 11: Qualitative comparison of top-4 images with and without LLM rephrasing. Some of the searching scenarios require cultural or knowledge context, LLM rephrasing helps extend the context.

The extension that LLM rephrasing provides can be divided as imagined elements and knowledge context, we analyze the cases in the following:

- **Imagined elements.** In Fig. 10, when user search "metallic mineral"or "Fluid mechanics sculpture", we may expect a shining and glaze surface. LLM rephrased results satisfy our expectation by adding our imagined styles into the query. In addition, our models may mix neural networks and website service networks, with adding the descriptions of the image, e.g. "with directed edges and nodes", the semantic accuracy can also be boosted. Another similar case is that models may mix virus and computer virus without extra details supplied.

- **Knowledge context.** Some of the objects need cultural or knowledge context. For example, as shown in Fig. 11, the understanding of abstract painting style and artsy style outfit. When searching cultural objects, LLM rephrased results seem more representative. The models are also sometimes puzzled to some of the expression without further explanation. For example, it may mix "girls of colors" and "girls paintings in different color", "a room in abstract paintings style" and "a room with abstract paintings" or "abstract painting of a room", "artsy outfit" and "art work on the outfit".

## D GPT-4V Judge Details

### D.1 Order consistency and GPT-4V win rate

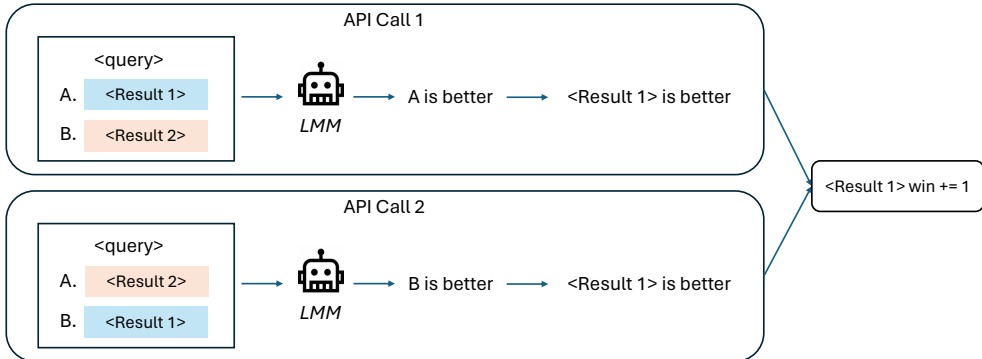

Figure 12: An example that result 1 wins. If two calls insist that result 2 wins, then the number of result 1 lose plus 1.

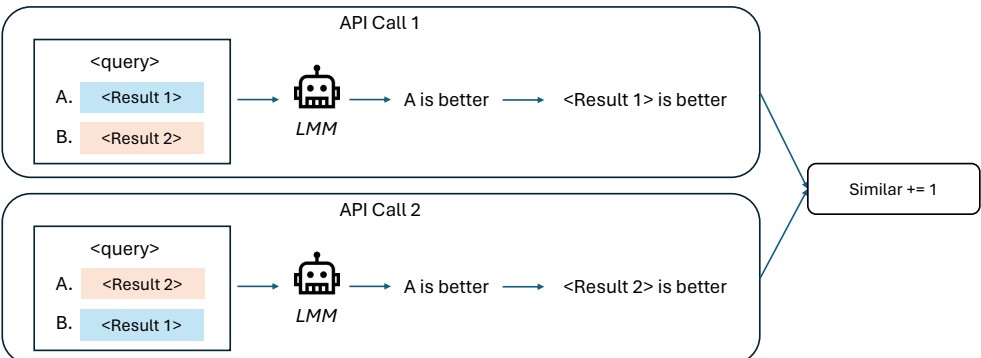

Figure 13: An example that two calls hold different idea, indicating that the results have few difference in quality, and we say these two results are similar.

**Order consistency (OC).** As shown in Fig. 12 and 13, for each query and results from R1 and R2, we call GPT-4V twice. If and only if the two results indicate a same winner, we commit that R1 wins or R2 wins (see Fig. 12). Otherwise we say they are similar and $N_s+=1$. Specific input and output of one call can be found at Appendix. G.1.

**Win rate.** When we are doing comparison experiments, we have to identify which one of the techniques is better. Thus using win rate is intuitional as if the win rate is bigger than 50%, system A should perform better.

**Win-and-similar rate.** This metric is more in line with the user when they compare two products. For example, If two products provide very similar results in most cases. In this situation, using win rate will be high-error and unreliable. In fact, high similar rate should yield that the user may use both two products with high probability. Therefore, we introduce win-and-similar rate to take similar cases into account, and we use this index when doing system level comparison.

## D.2 Rationality of GPT-4V judge

While GPT-4V exhibits strong capabilities in various vision tasks, it is still doubtful whether GPT-4V aligns well with human judgement on such subjective tasks. Empirical validation of this methodology is necessary. Therefore, we exploit our evaluation dataset, HPIR, to analyze the degree of consistency between GPT-4V's assessments and those of human labelers.

We directly compare the two groups (A and B) in HPIR using GPT-4V, and evaluate three different types of prompts, namely <ranker>, <scorer>, and <cp-scorer>. The <ranker> prompt involves merging ten images from group A and B into a single composite image (2 rows), upon which GPT-4V is tasked with determining the better row. The <scorer> prompt entails providing GPT-4V with a set of scoring guidelines to rate each group of images, where the group with the highest average score is deemed the winner. Lastly, the <cp-scorer> prompt requires merging ten images from group A and B into a single composite image too, then assigns scores to two groups concurrently. The winner is subsequently selected based on the higher scores obtained. Additionally, we also consider the order-consistency (OC) strategy discussed in Sec. 3. Given that the <scorer> approach evaluates a system independently, this method is free from order-consistency concerns. Furthermore, we engage five human experts who dedicate time to scoring for assessment purposes. The detailed descriptions of prompts are provided in supplementary materials.

Table 10: Evaluation of GPT-4V with different prompts (<ranker>, <scorer>, and <cp-scorer>) on HPIR. With order-consistency (OC), the <ranker> performs the best.

| HPIR Metric (%) | <ranker> | | <cp-scorer> | | <scorer> | Human experts | |
|---|---|---|---|---|---|---|---|
| | w/o OC | w/ OC | w/o OC | w/ OC | | w/o OC | w/ OC |
| Accuracy | 72.01 | 86.01 | 69.37 | 76.84 | 78.81 | 85.47 | 87.79 |
| Aesthetic | 64.48 | 86.70 | 60.94 | 80.79 | 77.83 | 82.52 | 85.00 |

The results are shown in Table 10. We observe that the <ranker> prompt with order-consistency performs the best among all the prompts, which is even comparable to human experts. This demonstrates the reliability of the GPT-4V judger. It is also evident that for pairs with minor aesthetic differences, GPT-4V's performance is considerably influenced by the order in which results are presented, mirroring a similar characteristic in human evaluators. Utilizing the order-consistency approach, which computes the win rate exclusively on consistent data, GPT-4V's evaluative accuracy is similar to that of humans. Given the subjective nature of aesthetic evaluation, the benchmarks set by humans on HPIR represent the ceiling for this data, indicating substantial potential for advancements in multimodal models.

## D.3 Details of Table 2

Table 11: Comprehensive details of Table 2.

| ID | Accuracy | | | | | Aesthetic | | | | |
|---|---|---|---|---|---|---|---|---|---|---|
| | A win | similar | A lose | win | win & similar | A win | similar | A lose | win | win & similar |
| 1 | 54 | 54 | 42 | 56.25% | 72.00% | 52 | 66 | 32 | 61.90% | 78.67% |
| 2 | 71 | 40 | 39 | 64.55% | 74.00% | 62 | 54 | 34 | 64.58% | 77.33% |
| 3 | 62 | 42 | 46 | 57.41% | 69.33% | 53 | 48 | 49 | 51.96% | 67.33% |
| 4 | 55 | 47 | 48 | 53.40% | 68.00% | 43 | 62 | 45 | 48.86% | 70.00% |
| 5 | 34 | 34 | 82 | 29.31% | 45.33% | 38 | 48 | 64 | 37.25% | 57.33% |
| 6 | 46 | 48 | 56 | 45.10% | 62.67% | 34 | 60 | 56 | 37.78% | 62.67% |
| Human labeler judge | | | | | | | | | | |
| 1' | 60 | 38 | 52 | 53.57% | 65.33% | 55 | 57 | 38 | 59.14% | 74.67% |
| 6' | 58 | 37 | 55 | 51.33% | 63.33% | 38 | 54 | 58 | 36.46% | 61.33% |

Table 12: Details of Table 3, origin query serves as baseline method.

| prompt | Accuracy | | | | | Aesthetic | | | | |
|---|---|---|---|---|---|---|---|---|---|---|
| | A win | similar | A lose | win | win & similar | A win | similar | A lose | win | win & similar |
| original | - | - | - | - | - | - | - | - | - | - |
| <detail> | 47 | 76 | 27 | 63.51% | 82.00% | 44 | 86 | 20 | 68.75% | 86.67% |
| <k list> | 36 | 83 | 41 | 53.73% | 79.33% | 38 | 94 | 18 | 67.86% | 88.00% |
| <kw dict> | 30 | 94 | 26 | 53.57% | 82.67% | 30 | 103 | 17 | 63.83% | 88.67% |
| repeat | 31 | 100 | 19 | 62.00% | 87.33% | 24 | 112 | 14 | 63.16% | 90.67% |
| <reorg> | 34 | 93 | 23 | 59.65% | 84.67% | 29 | 105 | 16 | 64.44% | 89.33% |

# E    Statistics of HPIR

In Table 13, we present HPIR results of our pretrained model, CLIP [38], and 3 aesthetic/quality models (CLIPIQA [50], MANIQA [62], IAP [42]). We further perform a simple model ensemble to enhance the capability of aesthetic assessment simply by adding the clip score and the scaled scores of aesthetic models. Although it is still different from 2-stage approach, we use this index to select a preferred aesthetic model as re-ranker in Sec. 2.3. This is because building a 2-stage retrieval system and searching scaling factors by calling LMMs to evaluate is unacceptable expansive. After grid search of models and scaling factors, we choose IAP [42] as the re-ranker.

Table 13: Performance of VL models and aesthetic models on HPIR benchmark.

| HPIR-Metric (%) | VL-model | | Image Aesthetic/Quality model | | | Ensemble |
|---|---|---|---|---|---|---|
| | Ours-PT | CLIP [38] | CLIPIQA [50] | IAP [42] | MANIQA [62] | CLIP + 1.25 IAP |
| Accuracy | 66.19 | 68.11 | 58.22 | 61.22 | 55.46 | 72.63 |
| Aesthetic | 59.36 | 62.05 | 58.35 | 63.32 | 53.21 | 67.99 |

It should be noted that it is unfeasible for the model to attain perfect scores across both metrics concurrently. Additionally, aesthetic models only provide an aesthetic score. In our HPIR metric calculation, we use this score to evaluate both metrics. Therefore, the accuracy evaluations for aesthetic models have no reference value. Moreover, since HPIR cannot evaluate the retrieval prowess of the model, we can only compare the HPIR metrics of the two models when they have comparable retrieval capabilities.

Fig. 14 shows an example query and its annotation file, each result is labeled with a golden label and a confidence. Fig. 15 displays the distribution of confidence scores with respect to two aspects (accuracy and aesthetic) in HPIR dataset. We observe a majority of queries with confidence scores between 0 and 0.5. Fig. 16 demonstrates the top-10 themes of the user queries with their frequencies. Because one query may have multiple themes, the sum of the count numbers can exceed 150. The most common themes are natural scene and human event, which follow the intuitive sense of human. Fig. 17 illustrates the interface of our labeling tool for labeling HPIR.

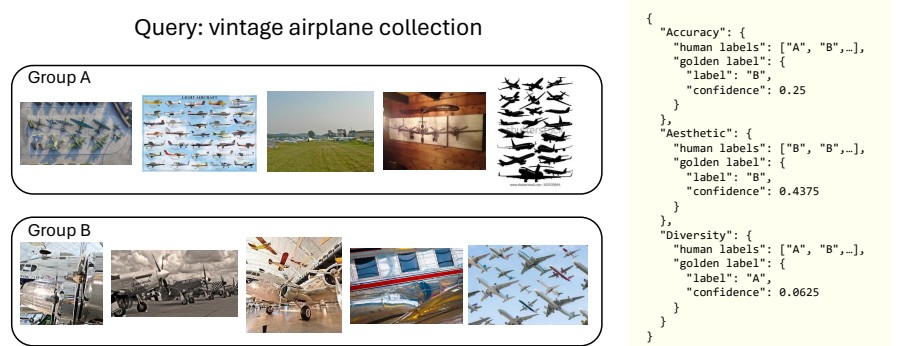

Figure 14: An example of human labeling for the query "vintage airplane collection."

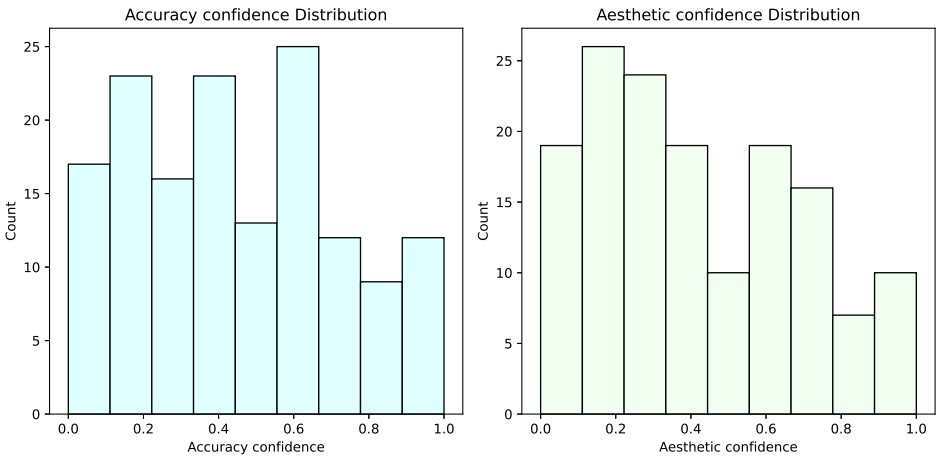

Figure 15: Distribution of confidence scores in HPIR dataset, with respect to three aspects (accuracy, aesthetic).

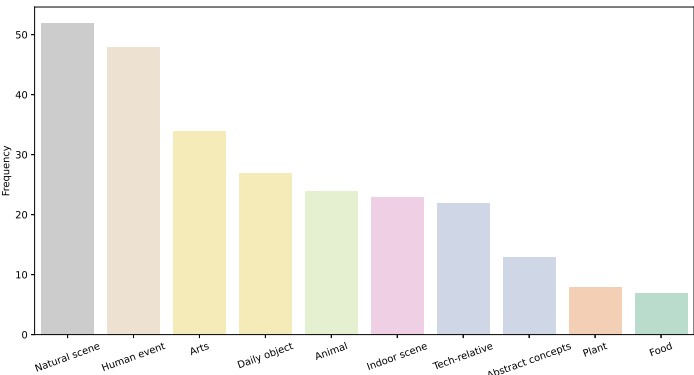

Figure 16: Top 10 themes of user queries and their frequencies. Note that one query may have multiple themes.

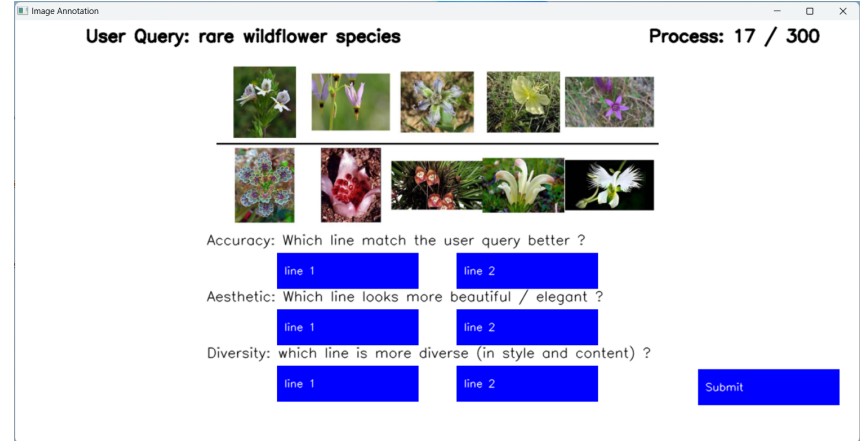

Figure 17: Screen shot of labeling tool for human labelers to label the HPIR dataset.

# F  LLM Rephrasing Prompts

## F.1  Prompt Template

Our utilized prompt structure is as follows:

```
You will be given a user query, and your task is to generate a
concise image description in English that aligns with the
user's intent. This description will facilitate the retrieval
of images that are both accurate and aesthetically pleasing
from the system. The description should be approximately
{n_words} words in length, constructed using the method
outlined below:

{method}

user query: {query}
```

Here, {method} indicates the rules that query should obey.

## F.2  Rephrasing Methods

We introduce the candidate methods of LLM rephrasing, the following items are possible substitutes for the method part of the template in the above template.

- <detail>:

  ```
  Supply visual details of the objects given in the user's query.
  ```

- <k list>:

  ```
  Generate a comma-separated list of succinct object descriptions,
  visual details, or stylistic elements to extend the aesthetic understanding
  and expectations, ordered from the most to the least significant.
  ```

- <kw dict>:

  ```
  Generate at least 2 key words for user query and then extend
  key word to description. Rank from most important key to
  least and follow this output format: <key>: <description>;
  <next key>:<description>; ...
  ```

- <reorg>:

  ```
  Repeat the original query in different linguistic
  organization for several times, splited by ","  and rank
  from the most accurate one.
  ```

We exemplify one query and various rephrasing results below, in Fig. 18-23.

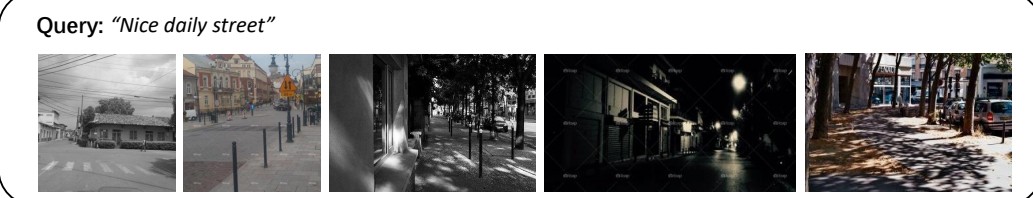

Query: *"Nice daily street"*

Figure 18: Original search query and its retrieved results.

Query: *"Nice daily street. Nice daily street. Nice daily street. Nice daily street. Nice daily street. Nice daily street. Nice daily street. Nice daily street. Nice daily street. Nice daily street."*

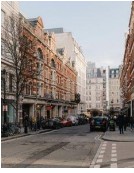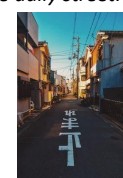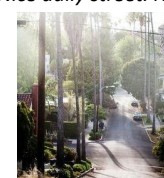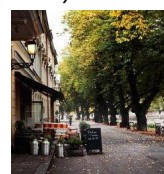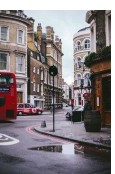

Figure 19: "repeat" method and its retrieved results.

**Query:** *"Daily street scene with a pleasant atmosphere, vibrant and inviting. A lovely urban landscape capturing the beauty of everyday life. Charming city street with a relaxed and welcoming ambiance. Picturesque daily street scene with a delightful and inviting vibe. A beautiful and serene city street on a peaceful day."*

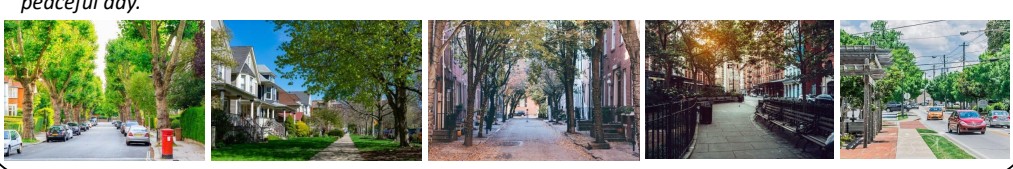

Figure 20: LLM rephrasing method: <reorg>, and its retrieved results.

**Query:** *"A serene, tree-lined street on a clear day. The image captures a peaceful urban scene with people strolling along the sidewalk, quaint shops and cafes lining the street, and colorful flower pots adding a touch of charm. The warm sunlight bathes the scene, creating a pleasant and inviting atmosphere."*

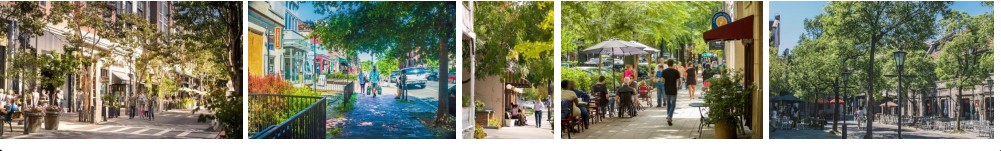

Figure 21: LLM rephrasing method: <detail>, and its retrieved results.

**Query:** *"People walking on a bustling city street, colorful storefronts with vibrant signs, urban skyline in the background, street vendors selling fresh produce, pedestrians in stylish and casual attire, outdoor cafes with sidewalk seating, bicycles and cars sharing the road, a relaxed and inviting atmosphere."*

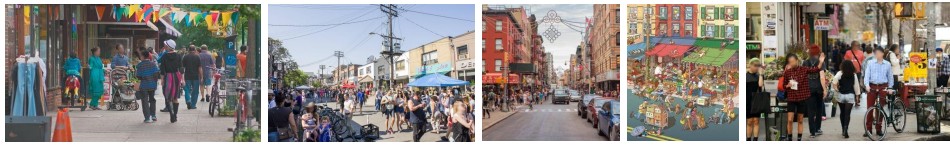

Figure 22: LLM rephrasing method: <k list>, and its retrieved results.

**Query:** *"Street: A bustling urban street with pedestrians and vehicles, capturing the vibrant daily life in a city; Daily: The routine activities of people going about their day, such as commuting, shopping, and socializing."*

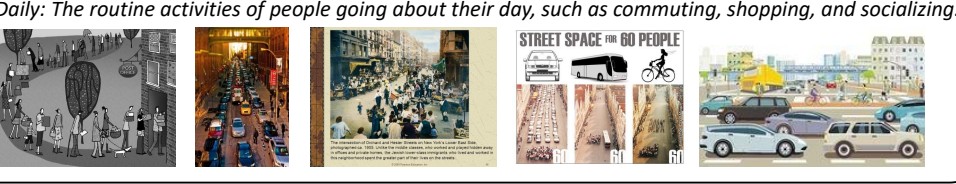

Figure 23: LLM rephrasing method: <kw dict>, and its retrieved results.

## F.3 Why Rephrasing Works

Our preliminary observation shows that there are 2 main reasons for this phenomenon.

- **Sequence length** is coupled with the image quality. As shown in Table 3 and Fig. 19, when we simply repeat the queries N times to extend the length of the query, the aesthetic performance increases stably. This is possibly because most of the vision-language models, including ours, CLIP and DataComp, are trained on crawled images from websites, where images with high quality or meaningful content are more likely to match a longer caption.

- **Visual element extension** continues to boost the aesthetic result. As shown in Table 3, using many of our rephrasing methods to extend the length of query can gain a better result comparing to repeat method. This prove the assumption that LLM rephrasing bring deeper visual understanding to queries and further boost the result.

There are more possible aspects, for example, query style can influence image style, saturation and even image shape. We believe there are more interesting inductive biases hidden within the data, and hence we hope these interesting phenomenons can encourage further researches into the underlying reasons.

# G  GPT-4V Judger Prompts

We introduce the details of GPT-4V judger in this section. We provide the system prompt (for GPT-4-vision model API) and example input and output for each.

## G.1  Method <ranker>

System prompt:

```
You are a judger of image retrieval systems whose job is to identify the system
that retrieve better images from a shared data base. You will be given a user
query, and an image containing 10 smaller images. In the provided image, there
will be 2 rows and each row will have 5 images, representing top 5 retrieval
images from 2 systems. We note that the first row is the result from <system 1>
and the second row is the result from <system 2>. Your judgement will be from 3
perspective: Accuracy, Aesthetic and Diversity.
Requirements:
**You MUST choose one even if you think two results are at same level**
Your output should be a JSON code that strictly follow this format:
```json
{
    "Accuracy analyze": <str: your analyze of how results of row 1 and row 2
    match the user query accurately, which row is better>,
    "Accuracy choice": <int: only 1 or 2, 1 for row 1 is better and 2 for
    the second row is better>,
    "Aesthetic analyze": <str: your aesthetic analyze of two row and which row
    is more beautiful and elegant>,
    "Aesthetic choice": <int: only 1 or 2, 1 for row 1 is better and 2 for the
    second row is better>,
    "Diversity analyze": <str: your separate analyze of how diverse of row 1
    and row 2 in content and style>,
    "Diversity choice": <int: only 1 or 2, 1 for row 1 is better and 2 for the
    second row is better>,
}
'''
```

Example input and output:

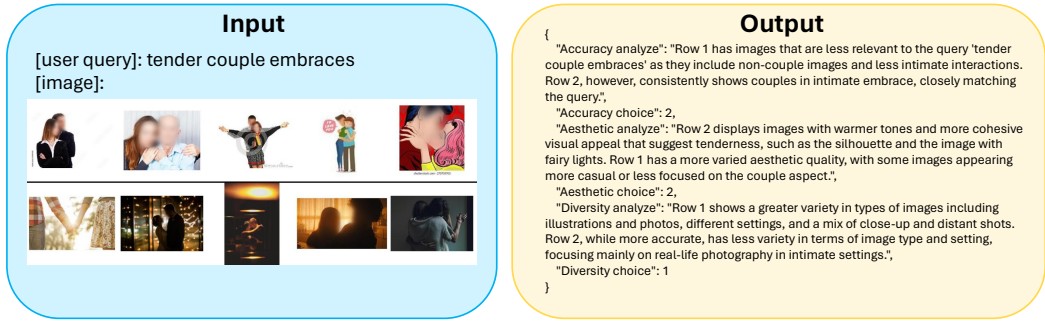

Figure 24: <ranker> (w/o OC) takes the input of two groups of images and prompts the model to choose the better one. In this example, line 2 wins the comparisons at accuracy and aesthetic aspects. In w/ OC setting, we take another call that exchanges line 1 and 2 of the input images.

## G.2  Method <scorer>

System prompt:

```
You are a judger of image retrieval systems whose job is to evaluate the system
by providing a score for its retrieval results of the given query. You will be
given a user query, and an image containing 5 smaller images, representing top 5
```

```
retrieval images from a system. Your judgement will be from 3 perspective:
Accuracy, Aesthetic and Diversity. You should provide a 1-5 score for each of
them.

Here is your scoring standard:
1 point: None of the result is satisfying, or even totally wrong.
2 point: more than 1 results are not acceptable, and the rest just make sense.
3 point: all results make sense, but only 1 or 2 are satisfying.
4 point: most of the results are satisfying, but 1 or 2 of them are not so outstanding
  or just make sense.
5 point: all results are very satisfying.

Your output should be a JSON code that strictly follow this format:
```json
{
    "Accuracy analyze": <str: your analyze of how the results match the user
    query accurately, then count the scores it can earn>,
    "Accuracy score": <int: one integer number from 1 to 5, representing the
    accuracy score>,
    "Aesthetic analyze": <str: your aesthetic analyze of how beautiful and
    elegant the results are, then count the scores it can earn>,
    "Aesthetic score": <int: one integer number from 1 to 5, representing the
    aesthetic score>,
    "Diversity analyze": <str: your analyze of how diverse of the results in
    content and style, then count the scores it can earn>,
    "Diversity score": <int: one integer number from 1 to 5, representing the
    diversity score>,
}
'''
```

Example input and output:

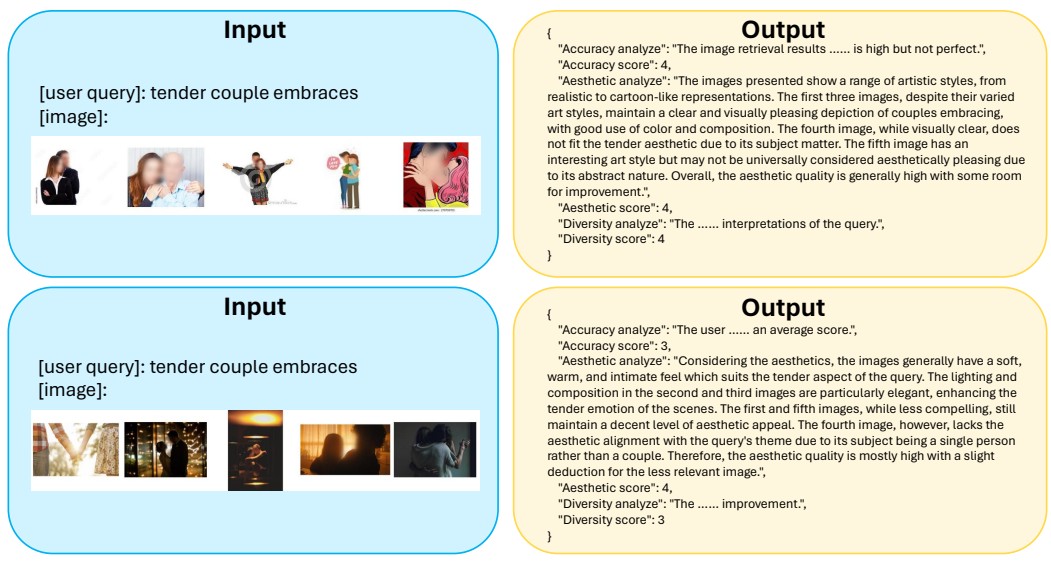

Figure 25: <scorer> requires two calls for two groups separately. We only leave the analysis of aesthetic aspect for visualization. In this example, the first group wins at accuracy and diversity aspects, and we will not take this sample into account when calculating aesthetic win-rate (as the two groups have equal scores).

## G.3   Method <cp-scorer>

System prompt:

```
You are a judger of image retrieval systems whose job is to evaluate the system
by providing a score for its retrieval results of the user's query. You will be
given a user query, and an image containing 10 smaller images. In the provided
image, there will be 2 rows and each row will have 5 images, representing top 5
retrieval images from 2 systems. We note that the first row is the result from
<system 1> and the second row is the result from <system 2>. Your judgement will
be from 3 perspective: Accuracy, Aesthetic and Diversity. Your job is to give a
score for both 2 systems from each of the above aspects.
```

```
Here is your scoring standard:
1 point: None of the result is satisfying, or even totally wrong.
2 point: more than 1 results are not acceptable, and the rest just make sense.
3 point: all results make sense, but only 1 or 2 are satisfying.
4 point: most of the results are satisfying, but 1 or 2 of them are not so
outstanding or just make sense.
5 point: all results are very satisfying.

Your output should be a JSON code that strictly follow this format:
```json
{
    "Accuracy analyze": <str: your analyze of how results of row 1 and row 2
    match the user query accurately, then count the scores they can earn>,
    "Accuracy scores": <List[int]: a list of 2 integers from 1 to 5,
    representing the accuracy scores of row  1 and row 2>,
    "Aesthetic analyze": <str: your aesthetic analyze of two rows of how
    beautiful and elegant they are, then count the scores they can earn>,
    "Aesthetic scores": <List[int]: a list of 2 integers from 1 to 5,
    representing the accuracy scores of row  1 and row 2>,
    "Diversity analyze": <str: your analyze of how diverse of row 1 and row 2
    in content and style, then count the scores they can earn>,
    "Diversity scores": <List[int]: a list of 2 integers from 1 to 5,
    representing the accuracy scores of row  1 and row 2>,
}
'''
```

Example input and output:

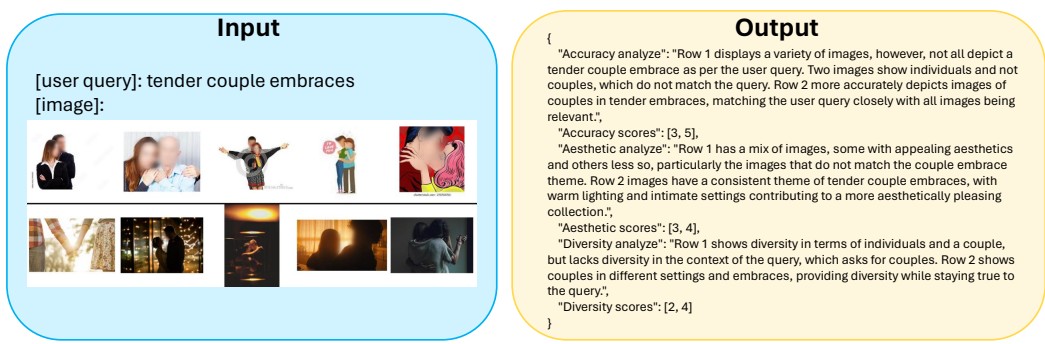

Figure 26: <cp-scorer> (w/o OC) takes the input of two groups of images and prompts the model to score them separately. In this example, line 2 wins all comparisons. In w/ OC setting, we take another call that exchanges line 1 and 2 of the input images and average the scores of two calls.

