# OpenReview forum: "Aligning Vision Models with Human Aesthetics in Retrieval: Benchmarks and Algorithms"
_NeurIPS.cc/2024/Conference — NeurIPS 2024 poster_

### Official Review · Reviewer_amTt · 2024-07-08

**Soundness:** 4
**Presentation:** 3
**Contribution:** 3
**Rating:** 7
**Confidence:** 5

**Summary:**

This paper studies the problem of aligning vision models with human aesthetic standards in a retrieval system. There are three key parts in the proposed model including LLM rephrasing, re-ranking, and RL fine-tuning. Two novel benchmarks are also introduced to integrate aesthetic quality into evaluation metrics. Experimental results demonstrate the effectiveness of the proposed method and the benchmarks.

**Strengths:**

1. This paper addresses the aesthetic quality issue in image retrieval systems and introduces a reinforcement learning fine-tuning strategy that enables the retrieval model to directly retrieve images based on both semantics and aesthetic quality, eliminating the need for multi-stage filtering. This approach holds significant value.
2. The paper introduces two evaluation benchmarks, addressing the limitation of current image retrieval benchmarks that fail to evaluate aesthetic quality.
3. The experiments are comprehensive, validating the importance of each component in the proposed method.

**Weaknesses:**

1. The methodological process described in the article is somewhat cumbersome, with Figure 2 merely outlining key processes and concepts in a rudimentary manner, thereby increasing the difficulty for readers to comprehend.
2. The authors appear to conflate "no-reference image quality assessment" with "image aesthetic quality assessment." While these tasks are indeed closely related, they are distinct. MANIQA, for instance, should not be regarded as an aesthetic quality assessment model, and its paper does not evaluate the model's performance on aesthetic datasets.
3. There remain some details in the article that are inadequately explained. It is peculiar that in Appendix Table 7, the same stride seemingly yields a different number of images.
4. The manuscript contains typos. For example, the indicator function symbol in Equation 11 is clearly garbled.

**Questions:**

1. In line 62, it is said that the open-source IAA datasets cannot be used for aesthetic retrieval evaluation. Can you give a further explanation?
2. In the first step of data preprocessing, the authors use the concept of "topic". Can you give a further explanation?

**Limitations:**

The paper does not discuss the limitations.

---

> ### Author Rebuttal · Authors · 2024-08-05
>
> ## W1: Cumbersome description of the method
> **R:** Thanks for your suggestion. We will find a way to further simplify the description of the method. The purpose of Fig. 2 is to illustrate the consistency of our approach and aesthetic concepts, and we will add more specific details to Fig. 3 that illustrates the specific steps. While at the same time, we will write a clearer index in the labels of Fig. 2 and Fig. 3 to avoid confusion for the reader.
>
> ## W2: Conflate "no-reference image quality assessment" with "image aesthetic quality assessment"
> **R:** Thank you! We will modify the description to distinguish between MANIQA and the aesthetic quality assessment model. We use this model because we have experimented with more image quality-related models that include these three in the paper and found MANIQA performs well.
>
> ## W3: Number of samples in Table 7
> **R:** Thanks! There may be some misunderstanding on the concept of stride, which refers to the interval between samples in our re-ordering sequence. The number of samples (as described in line 158-169, Sec. 2.3) is determined in terms of $u$ and $v$. The number of sampled images is $uv$ and the equation $|\mathcal{D}_{po}| = uC_v^2+vC_u^2$ shows the number of comparison pairs.
>
> For example, given a sequence of 100 images with $u$=$v$=5 and stride=2, we will sample $uv=25$ images with 2 as interval: [1,3,5,...,49]. If stride=3, it will be [1,4,7...,73]. The number of selected images is irrelevant to the stride.
>
> $|D_{po}|$ is the number of valid comparison pairs. When $u$=$v$=5 and stride=2 and we sampled [1,3,5...,49], we first split it with semantic dimension to [[1,3,5,7,9], [11,13,...],...,[...,47,49]]. There are 5 sequences of 5 images. For each sequence, such as [1,3,5,7,9], it contains $C_5^2$ valid comparison pairs: [(1,3), (1,5), ..., (7,9)]. Thus it will contain $5C_5^2$ pairs for all 5 sequences. Similarly, when we split with aesthetic dimension and result in [[1,11,21,31,41], [3,13,…]…, […,39,49]], it will also contain $5C_5^2$ pairs. Thus in this case $|\mathcal{D}_{po}|=5C_5^2+5C_5^2=100$.
>
> ## W4: Typos
> **R:** Thanks! We will fix the typos in the final version.
>
>
> ## Q1: Explanation on why IAA dataset cannot be used for aesthetic retrieval evaluation
> **R:** A data item in IAA can be formulated as `(Img, Text, Score)`, for example, `(<cat img>, cat, 0.6)`; and `(<dog img>, dog, 0.73)`. To be fair, retrieval must be compared with the same query, therefore our required data should be in the following format: `(Text, [{img_1, score_1}, {img_2, score_2}...])`; for example, `(dog,[{<dog img1>, 0.76}, {<dog img2>, 0.54}…])`. An alternative way is to retrieve images based on the IAA's queries, but we still need human resources to label the comparison. In addition, the queries' distribution does not match to the distribution of retrieval system's users. Thus, we decided to directly label the testing set.
>
> ## Q2: Explanation on "topic" in data preprocessing
> **R:** We expect that the distribution of the test set resembles the distribution of queries that are commonly searched by target users. User queries often involve user privacy that most companies will protect and cannot be directly obtained. A common alternative is to count the topic distribution, which are tags like "financial scene" or "indoor decoration" for queries. We generate queries based on the distribution of these topics to ensure that our query distribution is consistent with that of target users.

---

> > ### Comment · Reviewer_amTt · 2024-08-13
> >
> > Thanks to the author's response. Most of my concerns are addressed. As my current rating is already the highest, l will maintain the original score.

---

> > > ### Author Response · Authors · 2024-08-13
> > >
> > > Thank you for your valuable comments and your recognition of our work! We will add the explanation in the rebuttal to the revised version.

---

### Official Review · Reviewer_hvvN · 2024-07-11

**Soundness:** 3
**Presentation:** 3
**Contribution:** 3
**Rating:** 5
**Confidence:** 4

**Summary:**

The paper looks into the alignment task for vision and language models within retrieval models where properties such as visual aesthetic comes to play. To achieve this, the paper collects some data to design a metric suitable for taking into account human aesthetic evaluation. And employs an RL-based technique to exploit the human opinion for better aligning the retrieved images with human aesthetic preferences.

**Strengths:**

* It is well-written paper
* The concept of aligning vision with aesthetic preferences is interesting and useful in some applications.
* The experiments are well-designed and quite convincing.
* It is interesting that LLM rephrasing could improve the quality of results

**Weaknesses:**

* The proposed metric could be elaborated better and maybe explained how the study ensured the metric is not designed under influence of the model.

**Questions:**

* The work has been focused on visual aesthetics, given the LLM rephrasing results, it may be beneficial to look into the sophistication of language parts and how that could correlate with the aesthetic of the image. Could that motivate higher level of language sophistication is also correlated with higher visual aesthetics?
* Could you elaborate how the RL-based approach could scale?

**Limitations:**

The paper does not discuss any limitations with respect to the proposed method.

---

> ### Author Rebuttal · Authors · 2024-08-05
>
> ## W: The design of the proposed metric
>
> **R:** Thanks.
> We proposed two metrics in the paper: HPIR weighted accuracy and win rate.
> 1. The construction of HPIR requires retrieving and filtering images according to the query, and then manually picking representative images to label. In order to exclude the influence of models during retrieving and filtering process, as depicted in Sec. 3.1, we leverage multiple retrieval systems (including Google, Getty, Bing etc.) and retrievers built upon various base models (CLIP, OpenCLIP, DataComp etc.) with different sizes and training data, to make it as comprehensive as possible. In addition, we merged results from these sources together and manually picked representative pairs. The performer does not know which model the data is coming from, and this process decoupled the data from the model's influence.
> 2. We use win rate to quantify the comparison between different systems, which is a fair metric for different models. Additionally, the retrieval database we use is confidential, and we use human labeler to ensure that the results of GPT-4V are not seriously biased, and we use such indicators in the end-to-end evaluation at the system level, making it robust and reliable.
>
> ## Q1: The correlation to the sophistication of language
> **R:** Yes, higher levels of language sophistication is also correlated with higher visual aesthetics. But, "higher level of language sophistication" is also coupled or correlated with "aesthetic expectations and details" in our paper. Like we found in Sec. 4.2, even using longer query brings increase to the results. Higher level of language sophistication is also correlated with higher visual aesthetics due to the inductive bias within pre-training data.
>
> However, a nice system should be user-friendly, without the need to write complex languages, and reduce the budget that comes with calling GPT. With our RL-finetuned model, users can achieve similar results with simple language as with complex language.
>
> ## Q2: The scaling ability of RL-based approach
> **R:** Similar to RLHF in LLM, the more essential factor for whether an RL algorithm can scale mainly comes from the feedback. As an entry point to research, our feedback comes from aesthetic models, and it has a limited scaling upper-bound. In real application, we can obtain human feedback by, for example, user click-through rate. Feedback like this will exhibit excellent scaling properties and contribute to the continuous improvement of model capabilities, but due to privacy and policy reasons, it is out of the scope of our work.

---

### Official Review · Reviewer_9cZ7 · 2024-07-11

**Soundness:** 3
**Presentation:** 3
**Contribution:** 3
**Rating:** 5
**Confidence:** 2

**Summary:**

This work aims to align vision models with human aesthetic standards in a retrieval system. To do this, the authors propose a preference-based reinforcement learning method that fine-tunes the vision models to distill the knowledge from both LLMs reasoning and the aesthetic models to better align the vision models with human aesthetics. The authors further propose a novel dataset named HPIR to benchmark the alignment with human aesthetics.

**Strengths:**

1.	The idea of aligning vision models with human aesthetics in retrieval is interesting. This work has potential applications in various real-life applications.
2.	The authors’ motivation of utilizing the reasoning ability of large language models (LLMs) to rephrase the search query and extend the aesthetic expectations is insightful.
3.	The paper is well-written and informative.
4.	The proposed dataset HPIR can be used by fellow researchers in the related fields.

**Weaknesses:**

I feel it can be further improved in the following ways.

1.	For benchmarking human preferences, it might be better to record down the human variance in their annotations. I understand the authors used multiple annotations to ensure robustness, but since aesthetics is a subjective concept, human variance itself tells something.
2.	Following point 1, I feel the work can be made more solid if it includes some human evaluation studies on the experimental results. For example, in Fig. 5, it does not seem so obvious to me on the respective enhancement with finetuning.

Without the above two points, I feel the paper has somewhat overclaimed the "alighing vision models with human aesthetics".

**Questions:**

1.	Have the authors considered human variance in aesthetics perception?
2.	Are the objective metrics enough for results evaluation? Have the authors considered using human studies to evaluate the results?

**Limitations:**

The authors did not indicate limitations in their paper, and mentioned that they will discuss it in future. I feel this paper has clear limitations such as the indication of human variance and the evaluation of the results.

---

> ### Author Rebuttal · Authors · 2024-08-05
>
> ## W1 & Q1: Human variance
> **R:** Thank you for the suggestion.
> We provide a metric 'confidence' for representing the robustness of the label.
> The confidence score means the degree of agreement among all annotators, rather than a value provided by labeler. It is calculated through Equation 10.
> This confidence score and variance have similar indicative meanings. For example, if a piece of data has the confidence score of 0.6, it means that 24 labelers hold the same choice and the other 6 labelers choose the opposite. Thus, $$Confidence=\frac{2\times24}{24+6}-1=0.6$$
> We used this indicator and named it as confidence score because when both choices are supported by half of the labelers, $Confidence=0$, and when all labelers agree with a choice, $Confidence=1$.
> The distributions of all confidence scores are shown in Fig. 15 of Appendix E.
>
> As you mentioned, "aesthetics is a subjective concept," we used the confidence as a weight (see Equation 11) to ensure that our accuracy was only calculated on less controversial comparisons. This ensures the robustness of the experiment and excludes the influence of subjectivity.
>
> Per your advice, we further calculate the variance for these labels. Let the positive choice be 1 and the negative choice be 0, and then the labeling results can be formulated as a sequence like [1,0,0,1,...]. It is easy to derive the following formula:
> $$\text{variance} = \frac{ 2N_{pos} N_{neg}}{( N_{pos}+ N_{neg})^2}.$$
> Here are the details of human variance of our labeled data, accompanied with confidence score:
>
> |  Confidence Score Range  |  Avg. Acc Confidence  |   Avg. Acc Variance  |  Avg. Aes Confidence  |  Avg. Aes Variance  |
> |:---:|:---:|:---:|:---:|:---:|
> |$0.0 \leq c \leq 0.2$| 0.108 | 0.492 | 0.106 | 0.492 |
> |$0.2 < c \leq 0.4$| 0.324 | 0.446 | 0.343 | 0.440 |
> |$0.4 < c \leq 0.6$| 0.490 | 0.379 | 0.506 | 0.371 |
> |$0.6 < c \leq 0.8$| 0.676 | 0.269 | 0.676 | 0.269 |
> |$0.8 < c \leq 1.0$| 0.950 | 0.047 | 0.966 | 0.032 |
> |All| 0.441 | 0.364 | 0.411 | 0.377 |
>
> Last but not least, perhaps our statement of confidence is bothering you because we didn't explain its similarity to variance. We'll modify the description to make it more explicit.
>
> ## W2 & Q2: Lack of human evaluation
> **R:** Thank you for the suggestion.
> In Table 2 on page 8 of the main paper, we have presented a user study (last two rows), where we let multiple human labelers judge the images retrieved from models w. and w/o. alignment (using the same queries). These labelers are expert search engine users.
> Users judged which retrieved results from system A and B were better.
> The first six rows of Table 2 are judged by GPT-4V.
>
> Our description of this part does not link it to the phrase "user study", causing confusion to the readers, and we will revise the description of this section.
>
> We supplement more user studies over the experiments in Table 2 here:
>
> |   System A   |   System B   |   A to B win & similar rate by users   |
> |:-----:|:-----:|:-----:|
> | Ours-FT / Datacomp-15M | Ours-PT / Datacomp-15M | Acc: 65.3 / Aes: 74.7 |
> | Ours-FT / Internal-8M | Ours-PT + Re-rank / Internal-8M | Acc: 66.9 / Aes: 71.4 |
> | Ours-FT / Internal-8M | Bing Search / web | Acc: 49.1 / Aes: 56.6 |
> | Ours-FT / Internal-8M | Getty Search / getty images | Acc: 63.3 / Aes: 61.3 |

---

> > ### Comment · Reviewer_9cZ7 · 2024-08-12
> >
> > Thanks authors for the responses. I think this paper has potential to be published given the minor revisions done. I would like to keep the current borderline accept rating.

---

> > > ### Author Response · Authors · 2024-08-12
> > >
> > > We appreciate your recognition of our work and thank you for your valuable comments. We commit to add the human variance and additional human evaluation in the revised version.

---

### Official Review · Reviewer_CcyJ · 2024-07-12

**Soundness:** 3
**Presentation:** 2
**Contribution:** 3
**Rating:** 6
**Confidence:** 3

**Summary:**

This paper aligns the vision models with human values by leveraging LLM for query rephrasing and introducing preference-based reinforcement learning. The paper also presents a novel dataset named HPIR to benchmark the alignment with human aesthetics.

**Strengths:**

This paper introduces a novel approach to align visual models with human aesthetics, combining LLM rewriting to enhance query understanding and using preference-based reinforcement learning to fine-tune the model. The paper is comprehensive in experiments and introduces the HPIR dataset for benchmarking. The paper is well-structured and the methods are clearly explained. Key concepts are well defined and the use of diagrams helps to effectively illustrate the results. And this paper improves the aesthetic quality of results in image retrieval by aligning visual models with human preferences. The proposed method and dataset provide valuable ideas for future research in this area.

**Weaknesses:**

[W1] This paper lacks a detailed user study to validate the actual effectiveness of the proposed method. Including a user study with different participants to evaluate the subjective improvement of aesthetic alignment could provide stronger evidence for the actual effectiveness of the method.
[W2] Placing the related work in Section 6 makes it difficult for readers to have a clear understanding of the problem domain and existing research results before reading the specific methods and experiments, which is not conducive to the coherence of the paper structure.

**Questions:**

Computational Cost: Can you elaborate on the computational costs and resource requirements of the reinforcement learning-based fine-tuning process? Are there any optimizations that could reduce the computational burden?
User Study: Have you conducted any user studies to validate the improvements in aesthetic alignment from a user perspective? If not, do you plan to include such studies in future work?

**Limitations:**

Exploring possible negative social impacts, such as implications for privacy or how the technology might be misused in unintended ways Research could benefit from deeper analysis of how biases in training data affect model outputs beyond just aesthetics, particularly with respect to cultural and demographic diversity.

---

> ### Author Rebuttal · Authors · 2024-08-05
>
> ## W1 & Q: Lack of user study
> **R:** In Table 2 on page 8 of the main paper, we have presented a user study (last two rows), where we let multiple human labelers judge the images retrieved from models w. and w/o. alignment (using the same queries). These labelers are expert search engine users.
> Users judged which retrieved results from system A and B were better.
> The first six rows of Table 2 are judged by GPT-4V.
>
> Our description of this part does not link it to the phrase "user study", causing confusion to the readers, and we will revise the description of this section.
>
> We supplement more user studies over the experiments in Table 2 here:
>
> |   System A   |   System B   |   A to B win & similar rate by users   |
> |:-----:|:-----:|:-----:|
> | Ours-FT / Datacomp-15M | Ours-PT / Datacomp-15M | Acc: 65.3 / Aes: 74.7 |
> | Ours-FT / Internal-8M | Ours-PT + Re-rank / Internal-8M | Acc: 66.9 / Aes: 71.4 |
> | Ours-FT / Internal-8M | Bing Search / web | Acc: 49.1 / Aes: 56.6 |
> | Ours-FT / Internal-8M | Getty Search / getty images | Acc: 63.3 / Aes: 61.3 |
>
>
>
> ## Q: Computational Cost
> **R:** The RL-finetuning process only used 4 NVIDIA A100 GPUs for fewer than 5 hours. If using smaller batch size or techniques like gradient accumulation, it can be trained within an acceptable time even using other consumer-grade GPUs with more than 4GB of memory. This is a small burden compared to most of today's work, including our pre-training phase.
>
>
>
> ## W2: Related work
> **R:** Thank you for your comments, and we will make adjustments to the related work so that readers can better understand the article.

---

### Decision · Program_Chairs · 2024-09-25

**Decision:**

Accept (poster)

**Comment:**

This submission proposes a method to improve language-based image retrieval in line with human aesthetic preferences. The method includes (i) rephrasing queries to elicit better retrieved results; and (ii) collecting preference data and using them for an RLHF-like LLM alignment.

Reviewers found the work:
1. Generally well-written, illustrated and structured
2. Well-motivated
3. Novel and with important contributions to the community, including the RLHF-like method, the prompt-rephrasing approach, and the proposed dataset.

There were concerns about lack of clarity in some areas which the reviewers felt were sufficiently addressed, and some misunderstandings about user study results which were resolved. In addition there were ethics concerns wrt consent and privacy of subjects and labellers in the collected dataset which were also satisfactorily addressed. Ultimately all 4 reviewers recommend acceptance and the AC sees no reason to override the unanimous recommendation.

The authors are asked to include the recommendations of the reviewers (including the ethics reviewer) in the camera ready version.